# Resolving coral temperature vulnerability through heat and cold bleaching thresholds
Yusuf C. El-Khaled [1] ✉, Francisca C. García [1], Neus Garcias-Bonet [1], Matteo Monti[1], Erika P. Santoro[1], Matilde Marques [2,3], Natalie Dunn[1], Tina Keller-Costa [2,3], Christian R. Voolstra [4] & Raquel S. Peixoto [1,5] ✉

Coral bleaching is most commonly associated with heat stress, while cold-water bleaching remains an underrecognized threat. Building upon the widely used ED50 metric for standardized heat tolerance, we introduce a new metric, cold ED50, to quantify cold bleaching thresholds. By comparing cold and heat ED50s, we define the temperature variability range of coral species. To achieve this, we used Coral Bleaching Automated Stress System (CBASS) assays to assess heat and cold temperature tolerance across three Red Sea scleractinian corals (*Acropora* sp., *Pocillopora favosa*, *Stylophora pistillata*) during peak summer and winter along with microbial profiling. *Acropora* sp. exhibited the highest heat ED50 (38.68 ± 0.39 °C) in summer, while *S. pistillata* had the lowest cold ED50 (15.63 ± 0.26 °C) in winter. Our results revealed species-specific bacterial communities, with *Endozoicomonadaceae* dominating across seasons. We show that bleaching thresholds are negatively correlated with the abundance of *Endozoicomonadaceae* in *Acropora* sp. during summer. Notably, coral recovery capabilities after extreme temperatures also vary between species. This dual temperature tolerance framework offers a more comprehensive assessment of coral resilience and vulnerability in a rapidly changing climate.

Coral bleaching events, defined as the breakdown of coral-algal symbiosis through the expulsion of algal symbionts[1], have increased in frequency and intensity over recent decades[2,3]. While primarily linked to heat stress[4–7], recent research highlights the disruption of nutrient cycling within corals as a critical mechanism driving the collapse of this symbiotic relationship under thermal stress[8]. The first documentations of coral mortality events date back to the 1870s[9], with growing numbers since the 1980s[10,11], reporting mass bleaching events causing a dramatic loss of live coral cover on a global scale, even in reefs that have been previously hypothesized to represent refuges[6,12–16]. Just recently, the fourth global bleaching event due to hot temperatures was officially confirmed by the National Oceanic and Atmospheric Administration (NOAA). These events happened in less than three decades (1998, 2010, 2015/16, and 2023/24)[2,3,17], and have been highly detrimental to the world's coral reefs, with their effects being cumulative and their frequency increasing over time[3].

While these bleaching events received a lot of scientific and media attention, and are indeed the main current threat for coral reefs worldwide[2,3], indications of bleaching due to cold temperatures are far less common, even though the first indicators date back to the early 1940s[18]. The impact of cold water temperatures on coral physiology and health can be similar or even more detrimental than thermal, i.e., elevated temperature stress[19–23]. More than one century ago, Mayer (1914)[24] reported that the exposure of the coral *Montastrea annularis* to temperatures below 14 °C for more than 9 h was lethal. Findings of another experiment showed that corals were unable to feed heterotrophically when exposed to temperatures below 16 °C[25]. Cold temperature-induced bleaching has been observed in multiple hard and soft coral species worldwide (Fig. 1, Supplementary Data 1), and can occur repetitively, causing mass mortality events as documented for the Gulf of California[22,26–29] or the Florida Keys[12,18,19,30–33]. Nevertheless, cold water temperatures that lead to

[1]Division of Biological and Environmental Science and Engineering (BESE), King Abdullah University of Science and Technology (KAUST), Thuwal, Saudi Arabia. [2]Institute for Bioengineering and Biosciences (iBB) and Institute for Health and Bioeconomy (i4HB), Instituto Superior Técnico (IST), University of Lisbon, Lisbon, Portugal. [3]Department of Bioengineering, Instituto Superior Técnico (IST), University of Lisbon, Lisbon, Portugal. [4]Department of Biology, University of Konstanz, Konstanz, Germany. [5]Computational Bioscience Research Center (CBRC), Division of Biological and Environmental Science and Engineering (BESE), King Abdullah University of Science and Technology (KAUST), Thuwal, Saudi Arabia. ✉e-mail: yusuf.khaled@kaust.edu.sa; raquel.peixoto@kaust.edu.sa

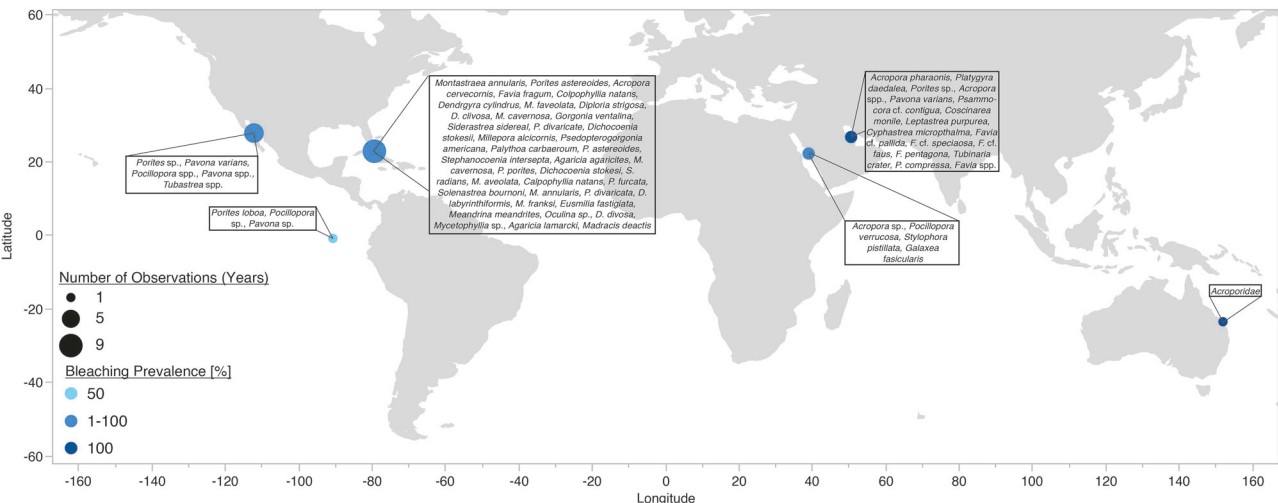

**Fig. 1 | Locations of historical bleaching events due to cold temperature stress reported in the literature.** Reports reveal that cold-temperature induced bleaching has been observed globally, and emphasize that cold-temperature related bleaching events may have not been observed and/or not reported. Details about cold bleaching observations can be found in Supplementary Data 1.

bleaching are rather unusual for tropical waters and are reportedly driven by two different phenomena.

Firstly, weather anomalies such as extremely cold winds can lead to a significant reduction of water temperatures termed cold spells, as reported for the Florida Keys and the Northern Bahamas[34,35], the Red Sea[36], the Persian Gulf[37,38] or the Great Barrier Reef[39]. The duration of these cold spells is highly dynamic and can last for days[35,39] or up to weeks and months[38]. Secondly, seawater temperatures can also vary significantly due to upwelling events or larger oceanic processes[40]. For example, a rapid temperature decline of 12 °C over six days was reported for the northern Galapagos Archipelago islands in March and May 2007, leading to water temperatures of ~16 °C that resulted in widespread bleaching as a result of this 'cold shock'[41,42]. Taken together, bleaching induced by cold water temperatures can be a 'silent', lethal threat on top of (and building on) heat stress during summer, even though global marine cold spells are potentially decreasing with global warming[43].

Many studies have determined thermal thresholds of multiple coral species around the world using the Coral Bleaching Automated Stress System (CBASS) or variations thereof to conduct acute thermal stress assays in a standardized manner[44–46]. For example, Evensen et al.[47] identified species-specific bleaching thresholds along a latitudinal gradient of the Red Sea. Coral health is often supported by a vast conglomerate of symbiotic microorganisms, that together with the coral host are referred to as the 'coral holobiont'[48–50]. In this context, the role of the microbiota (i.e., the host-associated microbial community and their interactions)[51] has been proposed to be fundamental for coral health[52–55]. Differences in the microbiome can be observed not only across species but also across genotypes of the same species[56–58], ultimately impacting coral health and bleaching susceptibility to heat stress[59–61].

Efforts in bacterial profiling are often linked to bleaching susceptibility as a response to heat stress[62,63]. The active enrichment of some bacterial species, for example, has been consistently associated with health improvements in corals (e.g. *Cobetia marina*, *Halomonas* sp., *Bacillus* sp.)[59,60,64]. Other prominent members across many coral species, including those in the Red Sea[65], such as *Endozoicomonas* spp. strains have been associated with stress[66] or considered putative symbionts of corals due to their presence in high abundances[67]. However, knowledge about the role of the microbiome contributing to coral resistance to cold-induced bleaching is lacking.

During the winters of 2023 and 2025, we observed paled and bleached *P. favosa*, *Acropora* sp., and *Echinopora* sp. coral colonies (Fig.1, Supplementary Data 1) when in-situ temperatures were lowest in the Central Red Sea. Additionally, bleached *Galaxea fascicularis* colonies were also observed in-situ in winter 2023. Despite a considerable number of reports on cold-bleaching showing that cold bleaching can occur globally affecting a considerable number of coral species (Fig. 1, Supplementary Data 1), empirical data remain sparse and fragmented, especially when compared to the wealth of research on heat-induced bleaching. To date, only two studies have reported putative cold water bleaching thresholds between 18 and 19 °C in *Acropora*, *Stylophora*[68], and *Pocillopora* morphospecies[28]. However, these thresholds were either based on field observations[28], or non-standardized long-term experiments with gradually decreasing temperatures[68], which limits their comparability and predictive power. A broader literature review (Supplementary Data 1) reveals bleaching associated with cold temperatures spanning a wide range (8-19 °C), underscoring the variability and complexity of cold-induced stress response in corals. Critically, standardized, experimentally derived cold-bleaching thresholds across multiple coral species remain lacking. To address this knowledge gap, we applied the Coral Bleaching Automated Stress System (CBASS) to empirically determine both heat and cold bleaching thresholds across three ecologically important Red Sea corals. By introducing a novel—to the best of our knowledge—cold effective dose (ED) 50, i.e., an empirically derived threshold when a decrease of 50% relative to the control is reached, and directly comparing it with heat ED50 values, we establish species-specific viable temperature ranges. In parallel, we characterize seasonal shifts in host-associated microbial communities to evaluate their potential roles in coral temperature resilience. This integrative approach offers essential insights into coral susceptibility across thermal extremes and contributes a vital framework for assessing coral vulnerability in the face of intensifying climate variability.

## Results

### Empirically derived bleaching thresholds

Photosynthetic efficiency ($F_v/F_m$) measurements were taken to assess the response of three coral species to both cold and thermal stress during two seasons using a series of CBASS assays. Classically, maximum monthly mean temperatures are used as a baseline reference serving as controls. For heat CBASS assays in summer, this resulted in heat CBASS temperatures of 31, 34, 37 and 40 °C, resulting in a reduction of $F_v/F_m$ of all coral species. To include a sufficient change in $F_v/F_m$ allowing the calculation of ED50 responses, the temperature range in heat CBASS assays in winter included 30, 33, 36 and 39 °C temperature profiles. Similarly, while the monthly minimum mean temperature in summer (30.5 °C) and its subsequent cold CBASS temperature profiles (30, 27, 24 and 21 °C) resulted in a decline of $F_v/F_m$ values allowing the determination of cold-

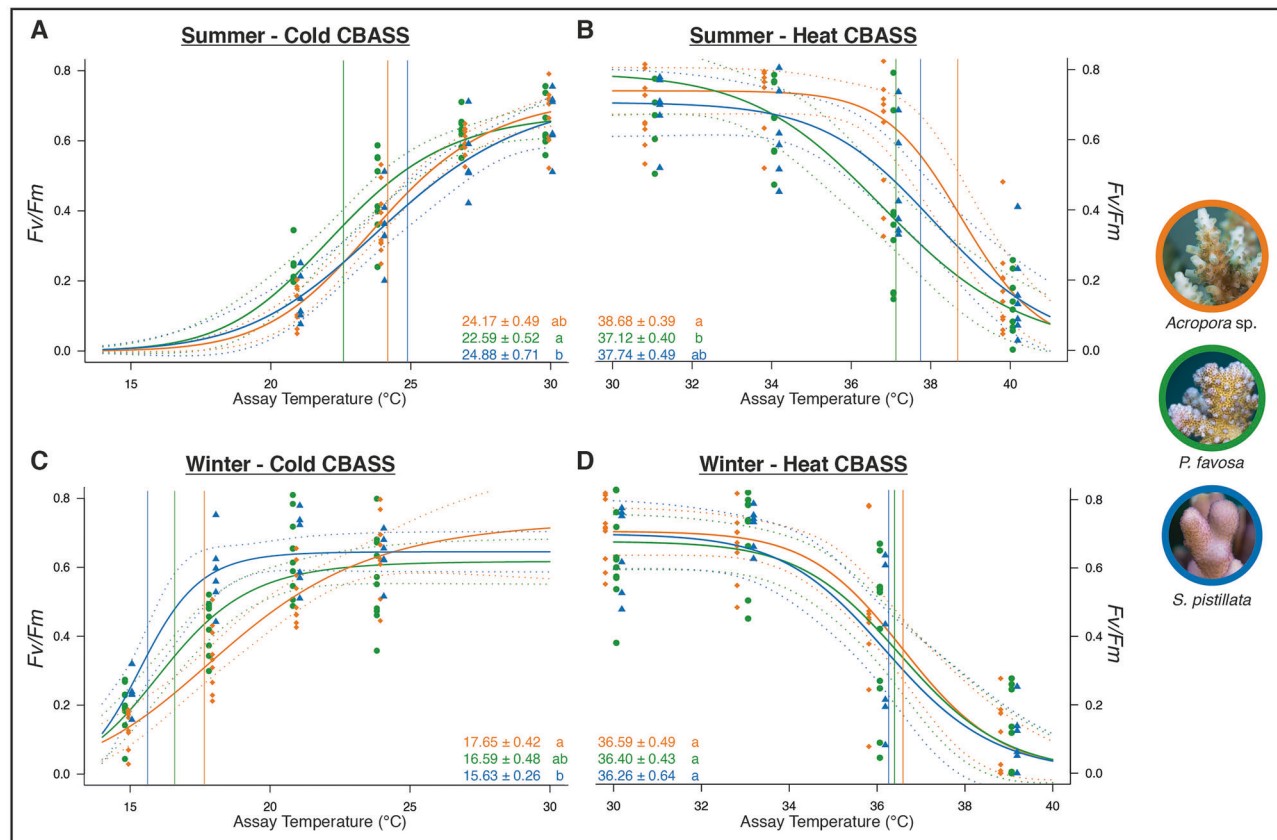

**Fig. 2 | Vital temperature range defined by respective temperature tolerance thresholds of *Acropora* sp., *Pocillopora favosa* and *Stylophora pistillata* in summer and winter.** Thermal tolerance thresholds (ED50) and changes in $F_v/F_m$ with 'cold' (**A**, **C**) and 'heat' (**B**, **D**) assay temperature profiles for each CBASS experiment performed in summer (**A**, **B**) and winter (**C**, **D**) using *Acropora* sp. (orange, rectangles, $n = 8$ biologically independent specimens in **A**, $n = 9$ in **B** and **C**; $n = 10$ in **D**, resp.), *P. favosa* (green; dots; $n = 8$ biologically independent specimens in **A**; n = 9 in **B**; $n = 10$ in **C** and **D**, resp.) and *S. pistillata* (blue, triangles, $n = 6$ biologically independent specimens in **A**, **C** and **D**, $n = 7$ in **B**, resp.). Dots, triangles and squares, respectively, represent the measured photosynthetic efficiency ($F_v/F_m$) at each experimental temperature for each CBASS assay and species. Lines reflect the log-logistic model fitted to each experiment (see "Methods") with dotted lines indicating the 95% confidence intervals of each log-logistic model. Color-coded vertical lines show the position of ED50 temperature thresholds matching the values in each panel of each species ± standard error. Different letters next to ED50 thresholds indicate statistical differences between species from the same assay ($p < 0.05$). Representative pictures of the coral species taken by Matteo Monti.

temperature bleaching threshold, we adjusted the temperature profiles for the cold CBASS assays during winter to achieve a quantifiable response.

Using these temperature profiles, the heat and cold temperature tolerance limits for three reef-building coral species (*Acropora* sp., *Pocillopora favosa*, and *Stylophora pistillata*) were determined during peak summer and winter temperatures (Supplementary Fig. 2). The upper temperature ED50 thresholds (hereafter: 'heat ED50') were highest for *Acropora* sp. (heat ED50 38.68 ± 0.39 °C), while *P. favosa* displayed lowest (heat ED50 37.12 ± 0.40 °C), whereas the heat ED50 for *S. pistillata* (37.74 ± 0.49 °C) ranged in between those of the other two species during summer (Fig. 2B). The heat ED50 threshold of *Acropora* sp. was significantly higher than that of *P. favosa* (Dunn test $p = 0.026$; see Supplementary Table 2), but in winter, all species exhibited similar heat ED50 thresholds (*Acropora* sp.: 36.59 ± 0.49 °C, *P. favosa*: 36.40 ± 0.43 °C, *S. pistillata*: 36.26 ± 0.64 °C; Fig. 2D).

A shift in the heat ED50 thresholds was observed between seasons, with *Acropora* sp. showing a ~ 2 °C decrease between summer and winter, and *P. favosa* exhibiting a ~ 1 °C decrease between seasons (Fig. 2B and D). The response to elevated temperatures varied among species in summer, as indicated by differences in thermal ED50 response curve shapes. *Acropora* sp. exhibited a heat decline width (*DW*) of 4.95 ± 0.87 °C (Supplementary Fig. 5 and Supplementary Table 3), lower than that of *P. favosa* (heat DW: 8.28 ± 1.88 °C) and *S. pistillata* (heat DW: 7.98 ± 2.34 °C). This trend

corresponded to the retention of $F_v/F_m$ values across a broad temperature range before a sharp decline above 37 °C, whereas *S. pistillata* and *P. favosa* showed a gradual decrease in $F_v/F_m$ values over increasing temperatures (Supplementary Fig. 5B and Supplementary Table 3). In winter, temperature-response patterns were less pronounced, with no visible species-specific differences and similar heat *DW* values among all species (*Acropora* sp.: 4.50 ± 1.05 °C, *P. favosa*: 4.44 ± 1.49 °C, *S. pistillata*: 5.40 ± 1.38 °C).

During winter, *S. pistillata* exhibited the highest tolerance to cold temperatures, with a lower temperature ED50 threshold (hereafter referred to as 'cold ED50') significantly lower than that of *Acropora* sp. (15.63 ± 0.26 °C vs. 17.65 ± 0.42 °C; Dunn test $p = 0.012$; Fig. 2C and Supplementary Table 2). The cold ED50 threshold for *P. favosa* (16.59 ± 0.48 °C) did not differ significantly from those of the other two species (Fig. 2C). In contrast, *S. pistillata* was the least cold-tolerant species during summer (cold ED50: 24.88 ± 0.71 °C) but showed the largest seasonal divergence in cold tolerance (cold *DW*: 9.25 ± 0.97 °C), exceeding the divergence of *Acropora* sp. (cold *DW*: 6.52 ± 0.91 °C) and *P. favosa* (cold *DW*: 6.00 ± 1.00 °C).

The response of photosynthetic efficiency to decreasing temperatures followed a similar trend across species in both summer and winter. All $F_v/F_m$ values decreased as temperatures declined, and the shape of the response curves was comparable to those observed for increasing temperatures in summer (Supplementary Fig. 5 and Supplementary Table 3). *S. pistillata*

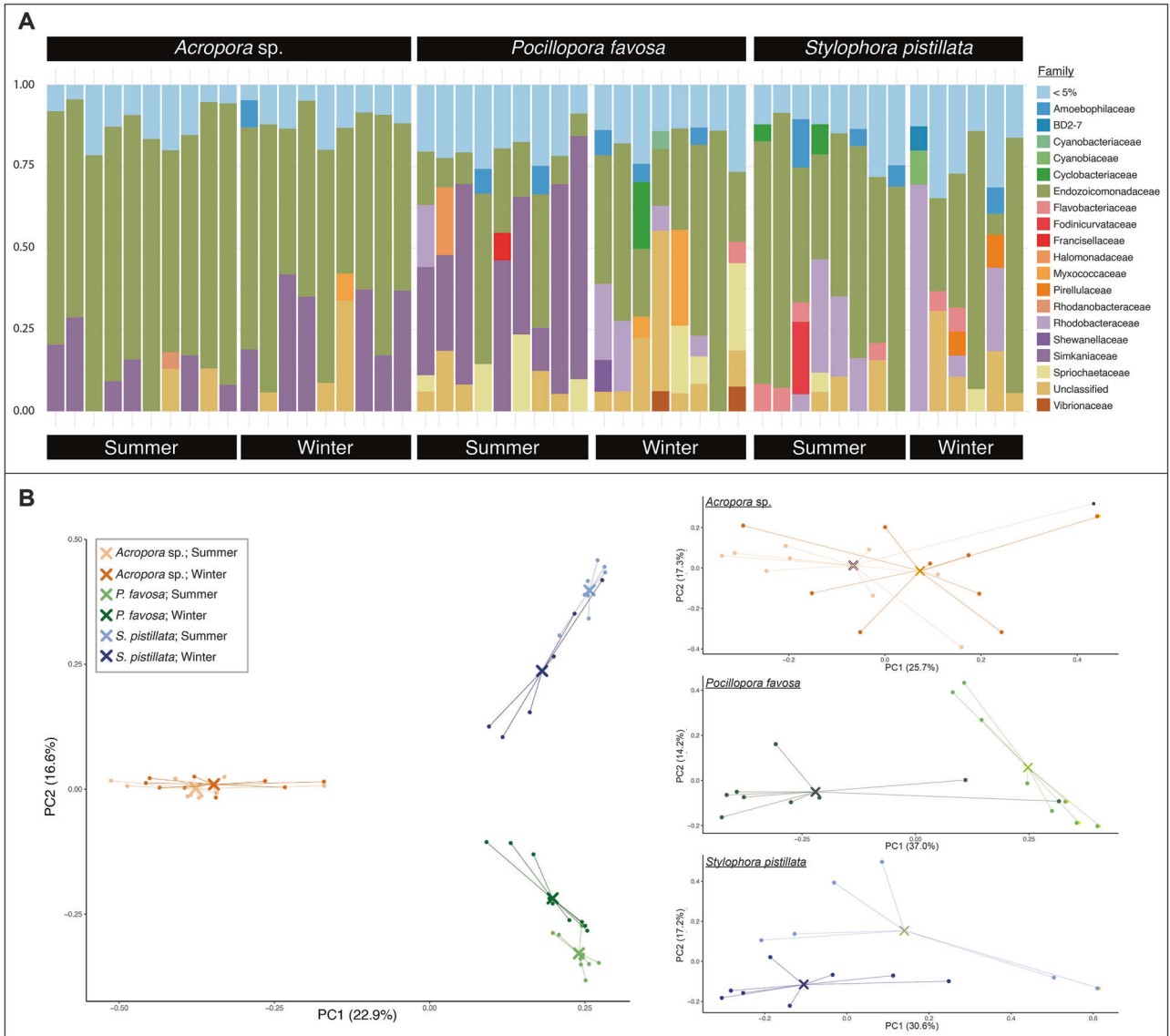

**Fig. 3 | Microbial profile of *Acropora* sp., *Pocillopora favosa* and *Stylophora pistillata* in summer and winter. A** Taxonomic composition of the bacterial communities of individual *Acropora* sp. ($n = 10$ and $n = 9$ biologically independent specimens, respectively), *P. favosa* ($n = 9$ and $n = 8$ biologically independent specimens, respectively) and *S. pistillata* ($n = 8$ and $n = 6$ biologically independent specimens, respectively) specimens during both seasons (summer and winter) on the family level (>5% abundance). **B** Principal coordinate analysis (PCoA) showing overall and within species differences of the coral-associated bacterial communities between the target species (*Acropora* sp. in orange, *P. favosa* in green, *S. pistillata* in blue) and seasons (summer in light colors, winter in dark colors). Note varying axes between plots.

maintained relatively stable $F_v/F_m$ values until a threshold temperature was reached, similar to the pattern observed for *Acropora* sp. in response to increasing temperatures. The corresponding cold *DW* values were lowest for *S. pistillata* (6.13 ± 2.29 °C), compared to *Acropora* sp. (11.59 ± 1.76 °C) and *P. favosa* (10.53 ± 2.31 °C).

## Host-associated bacterial profiling

We assessed the bacterial communities associated with all three coral species during both summer and winter. Raw reads from 50 coral fragments (3 species x 2 seasons x 6-10 colonies) were assigned to a total of 19,537 ASVs after quality filtering and demultiplexing. Subsequently, singletons were removed, as well as reads classified as belonging to chloroplasts, mitochondria, archaea, eukaryote and host contaminants, resulting in a total number of 16,465 ASVs. Note that the final dataset used for further analyses did not include non-classified ASVs at 'Kingdom' or 'Phylum' level. Finally, rarefaction at 100,000 reads, resulted in the loss of 3,687 ASVs leaving a final

total number of 12,778 ASVs across 50 coral samples. Analysis of bacterial communities associated with the three coral species across seasons revealed significant differences based on host species and sampling season (Bray-Curtis Adonis (species), $R^2 = 0.369$, df = 2, F = 15.318, Pr (>F) = 0.001; Bray-Curtis Adonis (sampling time), $R^2 = 0.035$, df = 1, F = 2.926, Pr (>F) = 0.004; Fig. 3B). *Acropora* sp. maintained a stable bacteriome across summer and winter (Bray-Curtis Adonis (*Acropora* sp.), $R^2 = 0.078$, df = 1, F = 1.446, Pr (>F) = 0.116), while significant shifts in bacterial community composition were observed in *P. favosa* (Bray-Curtis Adonis (*P. favosa*), $R^2 = 0.244$, df = 1, F = 4.852, Pr (>F) = 0.005) and *S. pistillata* (Bray-Curtis Adonis (*S. pistillata*), $R^2 = 0.149$, df = 1, F = 2.093, Pr (>F) = 0.021).

Despite these seasonal shifts, all three coral species maintained bacterial communities dominated by *Endozoicomonadaceae* (Fig. 3A), with relative abundances ranging from 20.72% (*P. favosa* in summer) to 74.98% (*Acropora* sp. in summer). Other abundant bacterial families were *Rhodobacteraceae* and *Simkaniaceae*. Seasonal differences in bacterial

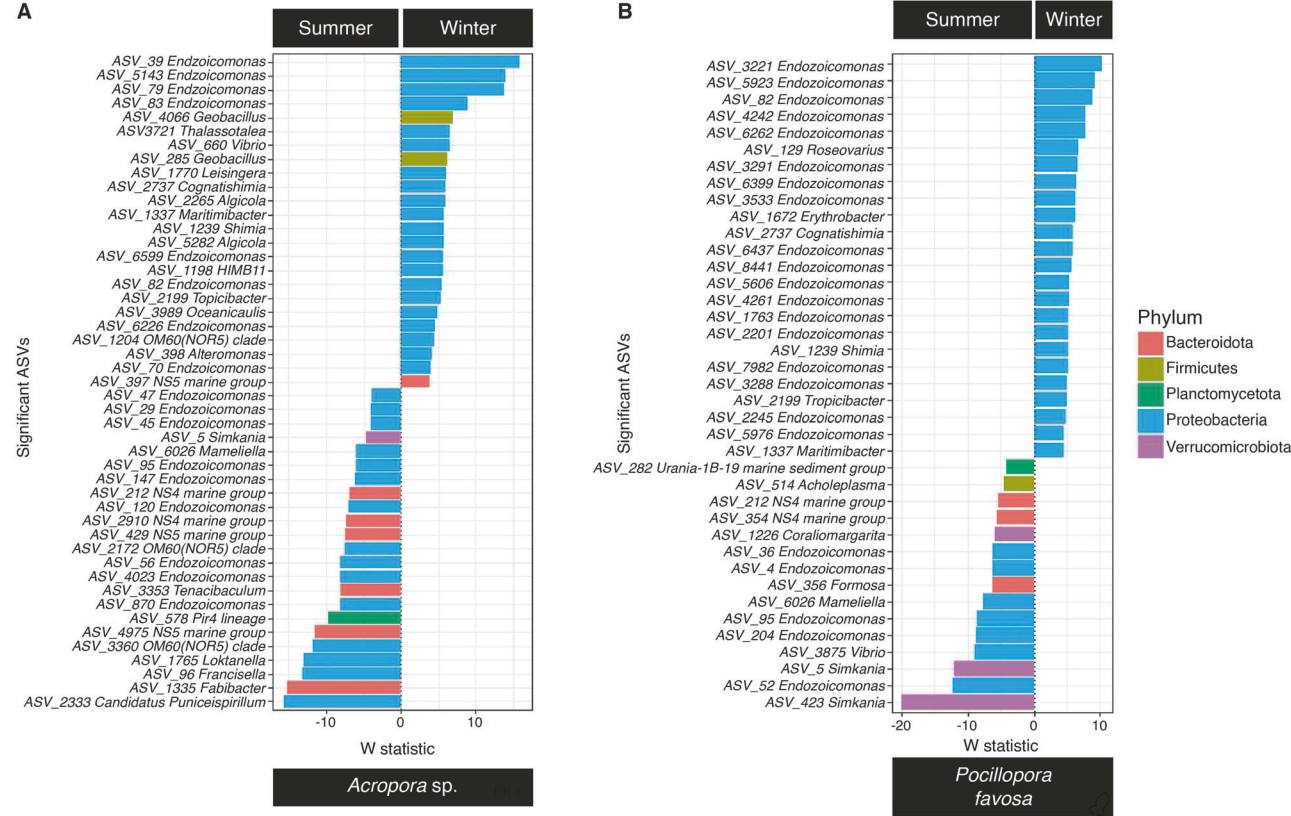

**Fig. 4 | Top differentially abundant ASVs with their associated genera.** Color-coded by phylum for **A** *Acropora* sp. and **B** *P. favosa*. No changes were detected for *S. pistillata*. The *W* test statistic from the ANCOM-BC2 is shown (negative *W* indicates decreased taxa while positive *W* indicates enriched taxa in winter compared to summer), with $n = 10$ and $n = 9$ biologically independent specimens, respectively for *Acropora* sp. in summer and winter, respectively; and $n = 9$ and $n = 8$ biologically independent specimens for *P. favosa* in summer and winter, respectively.

composition were particularly evident in *P. favosa*, where the relative abundance of *Endozoicomonas* increased significantly in winter (ANOVA, df = 1, F = 4.408, $p = 0.0531$). However, no significant seasonal differences were detected at the ASV level in *S. pistillata*, despite overall shifts in community composition. Differential abundance analysis revealed that 69 ASVs changed significantly between summer and winter in *Acropora* sp. (43 enriched in winter, 26 enriched in summer; Fig. 4A), while 49 ASVs changed in *P. favosa* (26 enriched in winter, 23 in summer; Fig. 4B). The top significantly enriched ASVs in *Acropora* sp. during winter belonged to the *Simkaniaceae* family ($n = 14$ ASVs). In *P. favosa*, 46.9% of differentially abundant ASVs between seasons belonged to *Endozoicomonadaceae*, followed by *Rhodobacteraceae* (16.3%). Calculated heat ED50 values showed a strong negative correlation with the abundance of *Endozoicomonadaceae* in *Acropora* sp. during summer (Spearman rank correlation: rho = −0.7090, $p = 0.0216$; Supplementary Table 1). Additionally, a positive correlation between cold ED50 values calculated for *S. pistillata* in winter and the abundance of bacteria belonging to *Rhodobacteraceae* approached statistical significance (Spearman rank correlation: rho = 0.7714, $p = 0.0724$; Supplementary Table 1), with a heterogenous distribution among the specimens.

### Recovery analysis

To assess recovery potential following temperature stress, photosynthetic efficiency was measured immediately after the heat stress phase ($t_7$) and after an 11-hour recovery period ($t_{18}$). Following exposure to extreme cold temperatures in winter (15 °C), a significant increase in photosynthetic efficiency was observed across all investigated coral species after the recovery phase (paired t-test $p_{BH-adj} = 0.02$ for *Acropora* sp., paired t-test $p_{BH-adj} = 0.037$ for *P. favosa*, paired t-test $p_{BH-adj} = 0.007$ for *S. pistillata*; Fig. 5B; Supplementary Data 2). During summer, no significant recovery of $F_v/F_m$ values was observed for the coldest treatment in any coral species (Fig. 5A), except for a significant increase in $F_v/F_m$ values only recorded for *Acropora* sp. after mild cold stress at 27 °C (paired t-test $p_{BH-adj} = 0.038$; Supplementary Fig. 7, Supplementary Data 2). In contrast, after exposure to extreme heat stress (40 °C), photosynthetic efficiency remained consistent or decreased across all species and both seasons (Supplementary Fig. 7, Supplementary Data 2).

### Discussion

Here we introduce and validate a metric, cold ED50, inferred from the commonly used short-term CBASS assay to empirically determine cold bleaching thresholds. This metric allows the resolution of species-specific viable temperature ranges by means of heat and cold standardized temperature tolerance thresholds (i.e., ED50s). We demonstrate that species-specific bacterial communities can vary significantly between seasons and across the three coral species. Additionally, we show that bleaching thresholds are negatively correlated to the abundance of *Endozoicomonadaceae* during summer in *Acropora* sp., and potentially with the abundance of *Rhodobacteraceae* during winter in *S. pistillata*, with each species displaying the peak in temperature tolerance in its respective season.

Adopted from the widely used ED50 standardized thermal tolerance thresholds (i.e., thermal ED50), we introduce 'cold ED50s' to define cold tolerance thresholds. Our modifications here for the cold CBASS were targeting the respective temperature profiles that were required to obtain an adequate response of the corals (i.e., decline in $F_v/F_m$) that allowed the calculation of corresponding cold ED50 temperature thresholds. While the classical CBASS temperature profiles are typically chosen based on local maximum monthly mean seawater temperatures of a given site or an

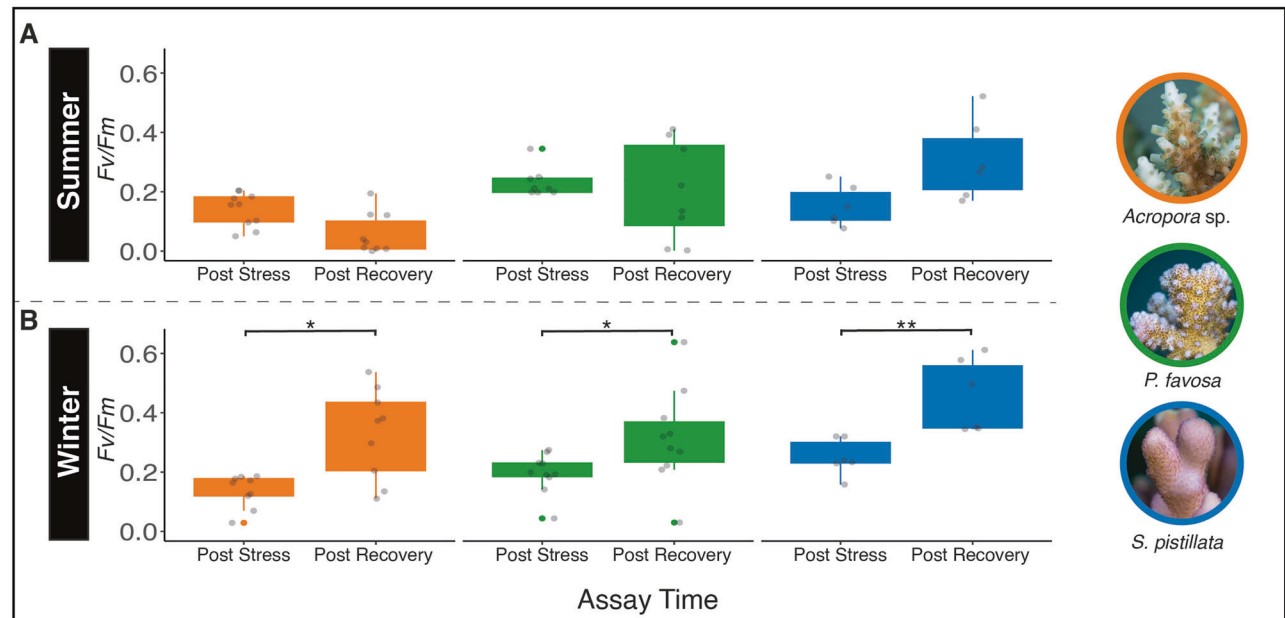

**Fig. 5 | Recovery capacities after extreme cold stress of *Acropora* sp., *Pocillopora favosa*, and *Stylophora pistillata*.** Photosynthetic efficiency ($F_v/F_m$) values obtained during 'cold CBASS' assay temperatures of 21 °C in summer and 15 °C in winter, measured immediately after the stress phase (t7) and an 11 h recovery phase (t18) in (**A**) summer v (**B**) winter for *Acropora* sp. (orange; $n = 9$ biologically independent specimens), *P. favosa* (green; $n = 8$ and 9 biologically independent specimens, respectively), and *S. pistillata* (blue, $n = 6$ biologically independent specimens). Boxplots show photosynthetic efficiency ($F_v/F_m$) of the three coral species, with boxes representing the interquartile range (IQR); whiskers extend to 1.5× IQR, and jittered points denote individual replicates. Asterisks indicate significant differences within species between measurement times based on paired t-test or Wilcoxon signed-rank test (* for $p < 0.05$; and ** for $p < 0.01$). Representative pictures of the coral species taken by Matteo Monti.

average temperature of multiple sites[46], we here adapted the profiles for both hot and cold CBASS according to the respective season (i.e., maximum mean summer and minimum mean winter, respectively) to achieve a decline in photosynthetic efficiency that allowed the calculation of corresponding ED50 values. This was done because of unchanging photosynthetic efficiencies performing heat CBASS assays in winter with temperature treatments using the monthly mean of 24.5 °C measured in a 30-day period before performing the CBASS assays. We hypothesized that corals of the Red Sea are well-adapted to elevated temperatures[69,70] with seasonal acclimations[71], with short-term stress of 34 °C not being sufficient to notably reducing the photosynthetic efficiency of the investigated coral species (Supplementary Fig. 1). Interestingly, using the minimum monthly mean temperature (i.e., minimum mean temperature 30 days before CBASS assays) in both summer and winter as a baseline for cold CBASS assays resulted in a significant decline of photosynthetic efficiency allowing the calculation of corresponding cold ED50 thresholds. Additionally, consistent with this seasonal framing, modelling studies from the Great Barrier Reef, Australia, show that anomalously warm, "hot winters", can precondition reefs and are associated with greater bleaching severity in the subsequent summer, for example during the 2016-2017 years[72]. Accordingly, prediction frameworks should explicitly incorporate winter-heat and prior-year heat metrics, not only summer-only exposure, because winter conditions help structure summer bleaching risk[72,73]. Therefore, we propose that determining the temperature tolerance range using both cold and heat CBASS assays as a critical tool complementing current CBASS assays allowing the determination of viable temperature ranges that comprise the entire spectrum, i.e., cold and hot temperature tolerance thresholds, of the coral species under investigation.

Heat ED50 thresholds during summer align with previous studies for the central Red Sea, with *Acropora* sp. showing the highest thresholds, followed by *S. pistillata* and *P. favosa*. Species-specific shifts between summer and winter ED50 thresholds (~2 °C *Acropora* sp.; ~1 °C *P. favosa*) likely reflect acclimation to seasonal temperature changes, consistent with previous findings (~3 °C for *P. favosa*, ~1 °C for *Acropora* sp.)[71]. Seasonal

differences for *P. favosa* also parallel observations from *P. damicornis* in the Great Barrier Reef, Australia, where a 1 °C difference in the bleaching tolerance across seasons has been reported[74].

In summer, the shape of heat ED50 response curves varied among species, suggesting species-specific reactions to elevated temperatures (sensu[75]). This variability aligns with differing DW values across species, potentially reflecting diverse thermal tolerances. *Acropora* sp. exhibited a similar response during summer and heat CBASS assays, maintaining thermal tolerance up to a critical threshold (sensu[75]). Combined with its lower DW compared to *P. favosa* and *S. pistillata*, *Acropora* sp. phenotypes appear comparatively tolerant[76], but not resilient[75]. Conversely, *S. pistillata* and *P. favosa* displayed response patterns with decreasing $F_v/F_m$ with stress intensity during summer and heat assays that correspond to previously detected patterns for the central and northern Red Sea[47,75]. The distinct response patterns in photosynthetic efficiency during heat assays observed in the present study were less pronounced during winter, with similar DW values observed across species (Supplementary Fig. 5). This indicates a link between long-term acclimatization or adaptation to naturally occurring elevated temperatures and species-specific response differentiation observed primarily in summer (sensu[75]). In this respect, while *P. favosa* shows a single genetic lineage across the entire Red Sea (except for the southern end), *S. pistillata* shows a putative pattern of regional genetic differentiation[77].

Our results also reveal that empirically derived cold ED50 thresholds are species-specific and exhibit significant seasonal variability (Fig. 2A, C), highlighting distinct coping strategies for high[71] and low temperature stress. Consistent declines in photosynthetic efficiency were noted across all species during cold CBASS assays in both seasons.

This study provides the first—to the best of our knowledge—experimentally determined cold bleaching thresholds, limiting direct comparisons with prior research. Notably, species-specific patterns identified in decreasing winter temperatures resemble those observed with increasing summer temperatures (Fig. 2C, B). Specifically, *S. pistillata* showed stable $F_v/F_m$ values until reaching a critical threshold,

analogous to *Acropora* sp. under summer heat stress (Fig. 2C, B). Conversely, *P. favosa* and particularly *Acropora* sp. exhibited declining photosynthetic efficiency correlated with cold stress intensity during winter, similar to *P. favosa* and *S. pistillata* responses to summer heat stress observed in the present (Fig. 2B) and a previous study[78]. Importantly, such species-specific cold response patterns were absent during summer, where $F_v/F_m$ values gradually decreased uniformly, similar to the response patterns to heat stress during winter. Our findings indicate that these species-specific response mechanisms manifest only during naturally elevated or lower seasonal temperatures, suggesting natural seasonal adaptation, and propose that similar adaptive mechanisms might exist for lower temperature extremes as suggested for high temperatures[71].

Currently, putative cold temperature induced bleaching thresholds are based solely on field observations[28]. Our experimentally derived cold ED50 thresholds are lower than those reported from in situ field observations in the Red Sea (sensu[46]). This discrepancy likely arises because short-term acute temperature-stress experiments like CBASS expose organisms to extreme stress rapidly, making them suitable for standardized threshold comparisons but not directly reflecting native thresholds, which typically are lower than those identified in heat CBASS assays[46]. As such, empirically derived ED50s do not reflect ecological bleaching temperatures but are designed to relatively compare coral responses to temperature stress in a standardized, empirical manner[46]. Moreover, this raises questions about whether corals from other locations within the Red Sea, particularly northern regions, or from elsewhere, might also exhibit lower cold bleaching thresholds due to the region's temperature gradients and whether species-specific differences within such gradients occur depending on genetic differentiation[77,79]. Our literature synthesis further supports this hypothesis, noting that 90% of coral bleaching events documented occurred at temperatures between 8 and 19 °C.

We demonstrate that the viable temperature range for corals varies with seasons, with cold stress ranges being extended by 6–9 °C in winter compared to summer, and heat stress ranges in summer being 1–2 °C above those in winter. This suggests natural acclimatization to naturally fluctuating temperatures with species-specific response patterns to both heat and cold temperature stress. Additionally, these findings complement that tropical reef-building corals live close to their upper thermal tolerance limits, such that even small (~1–3 °C) summer anomalies can trigger bleaching[80,81].

Given that coral thermal thresholds vary across geographic regions and across species[47] and are linked to bacterial community patterns[62], we characterized the bacterial communities associated with three coral species across summer and winter seasons. This approach allowed us to determine host-associated bacterial community dynamics potentially linked to seasonal temperature variations. The bacteriomes remained remarkably stable for *Acropora* sp. between seasons, whereas considerable shifts in the bacterial community composition were observed for *P. favosa* and *S. pistillata*. This stable bacterial community for *Acropora* sp. corresponds with highest heat tolerance in summer, but also highest cold ED in winter (i.e., least tolerant species in winter to cold temperatures). Our findings for *Acropora* sp. contrast previous research suggesting that corals belonging to the genera *Acropora* are generally harboring rather variable microbial associations[62,82,83]. Here, findings suggest that differences between summer and winter conditions did not impact the *Acropora* sp.-associated microbial composition on the family level, as reported in Epstein et al. [84] who did not identify a link between microbiome temporal variation and seasonality. Potentially, seasonal restructuring of coral microbiomes may in part reflect temperature-dependent growth optima of bacterial taxa, as seawater temperature is a major driver of microbial community dynamics[62,85]. However, changes in the bacterial community associated with *Acropora* sp. may still be induced by other factors than seasonality, as demonstrated in Ziegler et al. [86] who inferred changes in the microbiome of *Acropora* sp. due to anthropogenic impacts.

Conversely, the associated bacteriome of *P. favosa*, previously described as putatively inflexible[86,87], presented a variable associated bacterial

community across the two seasons. These results align with previous findings by Haydon et al. [88], showing that bacterial communities associated with other *Pocillopora* species are strongly influenced by environmental conditions. In this context, the observed differences in the bacteriome of *P. favosa* observed across seasons could indicate responses to in-situ temperature stress at the time point of sampling, even though both the donor colonies in situ and fragments used in the summer CBASS assays did not show any visual signs of bleaching or observable stress as highlighted by regular $F_v/F_m$ values in control CBASS assays. Potentially, such shifts in the coral microbiome may also be influenced by the surrounding seawater microbial community, which was not assessed in this study. Previous work has shown that environmental microbiota can seed or shape coral-associated bacterial assemblages[89]. Thus, part of the variation we observed may reflect concurrent changes in the ambient seawater microbiome with species-specific responses.

Corals of the genera *Stylophora* have been extensively investigated[36,47,69,90,91], with recent findings revealing distinct response patterns across the Red Sea, suggesting differing mechanisms to thermal tolerance[75,92]. Here, we show that *Stylophora*-associated bacterial communities vary across seasons, suggesting that this variability may have an important role in the coldest stress tolerant species examined in this study. However, provided functional traits along with candidate isolation need to be determined in future studies. Potentially, shifts in the bacteriome of *S. pistillata* may be driven by the extreme environmental fluctuations typical of the reef flats of the central Red Sea, where this scleractinian species is predominantly found[36,91]. Corals flourishing in these shallow and dynamic habitats may harbor a microbiome that adapts to changing environmental conditions, leading to a competitive advantage over other species.

As mentioned above, despite the species-specific seasonal differences highlighted in this study, the bacterial communities associated with all three scleractinian species were highly structured (i.e., few ASVs make up the majority of bacterial abundance). We found bacterial communities being mostly dominated by the family *Endozoicomonadaceae*, except for *P. favosa* during summer, with a relative abundance ranging between 20.72% (*P. favosa* in summer) and 74.98% (*Acropora* sp. in summer). *Endozoicomonadaceae* are Gammaproteobacteria that have been commonly observed in scleractinians and octocorals[67,93–97] with high abundances in coral species of the family *Pocilloporidae*[87], including those in the Red Sea[65]. Notably, while being present in all investigated specimens of *Acropora* sp., some specimens of *P. favosa* and *S. pistillata* show a relative paucity or even complete absence of *Endozoicomonas*, whereas it dominates the microbiome of conspecifics from the same site and season. Despite a considerable number of studies, a definite role and an understanding of *Endozoicomonas* at large is still missing[55,66]. For example, recent studies have observed ambiguous responses of *Endozoicomonas* abundances to stress with decreasing abundances[66,96,98–101] or quasi-invariant associations[86,87]. Their putative function as dimethylsulfoniopropionate (DMSP) degraders may be of particular importance, as DMSP forms dimethylsulfide (DMS) during heat stress, which in turn decreases oxygen free radicals, potentially mitigating coral bleaching[102]. Besides, DMSP degradation has been proposed[54] and validated[60] as a beneficial mechanism for corals provided by their bacterial symbionts. Accordingly, lower numbers of *Endozoicomonas* in *P. favosa* observed during summer could indicate responses of the corals' microbiome to in-situ temperature stress at the time point of sampling, despite the absence of physiological signs of stress and may have contributed to the significantly lower bleaching thresholds to heat stress during summer.

Interestingly, a strong, significantly negative correlation between the abundance of *Endozoicomonadaceae* during summer in *Acropora* sp. and ED50 values suggests that this bacterial family is connected to the heat tolerance of this species, particularly as *Acropora* sp. exhibited the highest heat ED50 in summer. This suggests a seasonal, species-specific microbiome restructuring towards an increase in abundance of bacterial groups that could play a critical role in coping with high temperatures, allowing the coral holobiont to withstand heat stress[83,86].

Although no significant seasonal shifts were detected at the family level in *Acropora* sp., we identified 69 ASVs that differed between seasons, compared to 49 in *P. favosa*. In contrast, *S. pistillata* showed no differentially abundant ASVs despite overall seasonal bacteriome changes. Among the ASVs enriched in winter for *Acropora* sp., the most significant belonged to the family *Simkaniaceae* (14 ASVs; Fig. 4A), a group within *Chlamydiia* commonly associated with corals. Despite Maire et al.[103] indicating the potential importance of *Simkaniaceae* for coral holobiont health as an important energy source for the coral host, their actual role remains still to be elucidated. A shuffling of *Endozoicomonas* ASVs in *Acropora* sp. and *P. favosa* (Fig. 4) has been previously documented in a fungid coral host species, where microbiome variations in *Ctenactis echinata* correspond with environmental factors such as sampling season, water quality, and substrate availability[104]. In the present study, specimens were collected from an area with relatively consistent water quality and substrate conditions. Given this, it is plausible that the observed shuffling of *Endozoicomonas* ASVs is influenced by the different sampling times[85] as well as host-specific factors[49,105].

Overall, we report species-specific dynamics related to coral-associated bacterial communities and connect these bacterial patterns with the corals' responses to temperature stress observed in CBASS assays. Specifically, we found that bleaching thresholds correlate with the abundance of *Endozoicomonadaceae* during summer in *Acropora* sp., and potentially with the abundance of *Rhodobacteraceae* during winter in *S. pistillata*, each representing the most tolerant coral species in their respective seasons. The potential connections between the seasonal shuffling of *Endozoicomonas* observed in *P. favosa* and *Acropora* sp., their less structured microbiomes, and their species-specific ED50 thresholds remain unclear and warrant further investigation.

Here, we assessed viable temperature ranges by determining the photosynthetic efficiency of three coral species using the classical and 'cold' CBASS assays over two seasons. CBASS assays have been designed with photosynthetic efficiency measurements being taken after the stress phase (i.e., $t_7$), and in some cases, after a subsequent overnight recovery period (i.e., $t_{18}$)[46]. While the $t_7$ measurements are used to resolve differences in thermal tolerances between coral populations[45,47,69,75], measurements at $t_{18}$ allow to assess differences in the response to temperature stress[92], hypothetically in the form of species-specific recovery potentials after temperature stress. Therefore, we here compared the photosynthetic efficiency measured after the stress phase ($t_7$) and a subsequent 11 h recovery phase ($t_{18}$; Fig. 5, Supplementary Fig. 7). Following heat stress in the hot CBASS assays, the photosynthetic efficiency remained similar or decreased for all species and specimens in both summer and winter. These findings correspond with a previous study that showed no significant changes in terms of bleaching or mortality patterns when incubated for longer periods (7 days) after heat stress compared to heat CBASS-style assays[106]. Overall, low photosynthetic efficiency values (~0.2) as observed in the present study across seasons and species after extreme heat stress indicate either the absence of zooxanthellae and/or severe damages of the photosystem II apparatus[107].

In contrast to heat stress, we observed a significant increase in the photosynthetic efficiency for all investigated coral species in winter, and for *S. pistillata* in summer following an 11 h recovery phase at control temperatures after being exposed to extremely cold temperatures (Fig. 5). These results indicate species-specific recovery potentials, highlighting that i) extreme heat stress impacts the photosynthetic system more severely than extreme cold stress, and/or ii) all coral species appear to possess mechanisms to cope with extremely low temperatures, expressed via an increased photosynthetic efficiency after a recovery phase subsequent to extremely cold stress (Fig. 5)[108]. Even though we did not assess potential structural damage to chloroplasts in the present study, Bieri et al.[108] reported that cold stress can impair chloroplast integrity and function in corals, suggesting that the enhanced photosynthetic efficiency we observed following a recovery phase may reflect compensatory mechanisms at the cellular or holobiont level rather than the absence of cold-induced damage. Such mechanisms may not only be linked to the host but to other members of the holobiont, as reflected

in the specific correlations between bacterial profiles and temperature thresholds observed in this study, and previously hypothesized[49,52,54,55,109]. Given the short duration of CBASS assays and observed recovery potentials, we hypothesize that these mechanisms seem highly dynamic, which reinforces the microbiome potential[110], due to the short generation times and quick adaptation and response to stress[111]. Linked to the predominant occurrence on reef flats of the central Red Sea, particularly *S. pistillata* experiences more extreme temperature fluctuations compared to corals growing deeper[36,45]. As such, corals flourishing in shallow habitats may have competitive advantages, as they experience a wider range of diel environmental conditions, such as extreme temperature fluctuations, occasionally reaching 18 °C in winter in the central Red Sea[36].

Determining heat and cold temperature tolerance thresholds, and recovery potentials, allows resolving the viable temperature range of corals, with implications for our understanding and prediction of climate change consequences in coral reefs. The results from our study highlight that all investigated coral species are prone to both cold and heat-stress induced bleaching, with measuring the viable temperature ranges of coral species offering an approach to assess climate resilient reefs. Importantly, combining hot and cold CBASS assays allows resolving the full viable temperature range of corals, highlighting that species with broader ranges may be more resilient to heat stress by withstanding and recovering during subsequent cold events[112]. Winter bleaching events triggered by severe weather anomalies or cold-water upwelling can pose threats to corals comparable to those caused by heat stress events. Given the increasing frequency of thermal bleaching in summer, cold stress represents an additional environmental challenge that could compromise periods critical for coral recovery[113]. In the light of the increasing occurrence of weather anomalies and climate change, corals that experienced temperature stress in summer may be more susceptible to cold stress in following winter months, or vice versa. Thus, the interplay between cold and heat-induced bleaching may significantly influence overall coral resilience and mortality. While coral responses to heat stress are extensively documented, cold-water bleaching remains a largely overlooked, 'silent', yet globally relevant threat. Considering the increasing occurrence of extreme temperature fluctuations due to climate change-driven anomalies, our findings suggest that certain coral species may exhibit higher recovery potentials, thereby gaining competitive advantages in adapting to these changing thermal conditions.

## Methods

### Study site, environmental background measurements, and bleaching observations during winters 2023 and 2025

The 'Coral Probiotic Village' (CPV), an underwater laboratory, located at the Northern Al Fahal reef complex in the central Red Sea (22.305 N, 38.966E)[114] was established in June 2021. The overarching purpose of the CPV is to provide a platform to investigate integrated and holistic approaches determining the potential of innovative solutions to rehabilitate coral reefs. Ongoing efforts in the CPV include the acquisition of physicochemical variables, such as seawater temperature, salinity, and currents, in a high temporal resolution (see ref.[114] for detailed information). To synthesize cold temperature induced bleaching events, we performed a narrative literature review[115] aiming to summarize information on such events, including relevant in-situ temperatures, impacted coral species, and the spatial and temporal distribution of such events. Results of the narrative literature review and the performed explorative bleaching surveys in the Central Red Sea in 2023 and 2025 are presented in Fig. 1 and Supplementary Data 1.

### CBASS acute thermal assays and physiological measurements

Coral fragments from the species *Acropora* sp., *Pocillopora favosa*, and *Stylophora pistillata* were collected from the CPV in August 2023 (hereafter: summer) and in February 2024 (hereafter: winter) at depths of 3−8 m. This study was conducted under the Institutional Biosafety and Bioethics Committee (IBEC) approval (23IBEC097) following the guidelines of the Kingdom of Saudi Arabia National Committee of Bioethics (KSA NCBE).

Coral colonies were tagged in the summer of 2023 to ensure that the same genotypes were monitored again in the winter of 2024. However, due to bleaching and subsequent mortality in the Red Sea during the summer of 2023[116,117], different *S. pistillata* colonies had to be sampled in February 2024. After collection, all coral fragments were immediately transferred to temporary aquaria filled with freshly on-site sampled reef water on the boat and kept at ambient light and temperatures. An extra fragment of each genotype was sampled for microbial analysis (see sections below). We started the experiments at the wet lab facility of the Coastal and Marine Resources (CMOR) Core Lab at the King Abdullah University of Science and Technology (KAUST) using the Coral Bleaching Automated Stress System (CBASS) less than three hours after sampling, using water collected from the reef site. During these three hours, coral fragments were kept under ambient temperatures aiming to bypass any possible acclimation and to avoid adjustments to possible tank conditions aiming to have close to in-situ conditions[45]. The CBASS was chosen to conduct replicable, identical, short-term temperature stress assays[46]. Full technical details of the CBASS setup can be found in refs. [45] and [46]. This assay was designed to rapidly capture biological stress responses of corals that are exposed to heat stress, using increasing temperatures (hereafter: 'heat CBASS') with a control temperature based on the maximum monthly mean temperature, and mild (+3 °C), moderate (+6 °C) and extreme (+9 °C) heat stress. The monthly mean of the daily maximum seawater temperature records of July 2023 (here: 31.5 °C) was chosen as control temperature for the classical CBASS setup here, resulting in temperature treatments of 31 °C, 34 °C, 37 °C and 40 °C. To assess the lower temperature thresholds, we inverted the classical CBASS temperature profiles, using the mean minimum temperature of July 2023 (here 30.5 °C) as a control, and applied the same temperature intervals, resulting in temperature treatments of 27 °C, 24 °C and 21 °C (hereafter: 'cold CBASS'). In February, we similarly chose the mean minimum temperature of January 2024 (24 °C, respectively; Supplementary Fig. 2) to apply the heat (24, 27, 30, 33 °C) and cold (24, 21, 18, 15 °C) stress temperature profiles. However, the initially chosen temperatures for the heat CBASS assay (24, 27, 30, 33 °C) did not alter the photosynthetic efficiency that allowed the calculation of bleaching thresholds (see results in Supplementary Fig. 1). We therefore chose a similar target temperature profile to match the heat CBASS assay in summer, i.e., 30, 33, 36 and 39 °C (Supplementary Fig. 3). In-situ temperatures for July 2023 and January 2024 from the same location are displayed in Supplementary Fig. 2. Coral fragments for both heat and cold CBASS assays were collected from the same set of tagged coral colonies (i.e., distinct but clonal fragments for each assay). For cold CBASS assays, coral fragments were collected on August 15th, 2023, and on February 14th, 2024; coral fragments for heat CBASS were collected on August 20th, 2023, and on February 18th, 2024, respectively. All temperature treatments were heated or cooled, respectively, over a 3 h period, held for another 3 h at the respective target temperatures, and then decreased or increased, respectively, back to the baseline temperature over 1 h. For the remaining 11 h of the total 18 h experiment, coral fragments were kept at the baseline temperature. Temperatures for each assay were monitored at a 1-minute interval using HOBO Pendant Onset temperature logger (see Supplementary Fig. 3).

### Coral bleaching assessment

Dark-acclimated photosynthetic efficiencies, i.e., the maximum photosystem (PS) II quantum yield ($F_v/F_m$), were measured to determine ED50-based temperature tolerance limits. The effective dose 50 (ED50) describes when a decrease of 50% relative to the control (baseline) value for each individual coral genotype within a population for a response metric (i.e., $F_v/F_m$ values, visual bleaching scores, etc.) is observed[46]. Photosynthetic efficiency was determined using a pulse-amplitude modulated fluorometer (Diving-PAM-II; Walz; blue-emitting light) by measuring the maximum quantum yield of PS II (*Fv/Fm*), which is a metric used to describe how efficiently absorbed light is converted into chemical energy by the symbiotic algae's photosynthetic machinery[118]. Photosynthetic efficiency serves as a non-invasive indicator of algal photo-physiology[119], and reflects the

performance of the symbionts' photosystem[120]. A decline in $F_v/F_m$ usually indicates photodamage, impaired electron transport, or protective downregulation in response to stress. Declines in $F_v/F_m$ often precede visible bleaching, with associated ED50 thresholds being used as thermal tolerance proxies informing bleaching susceptibility[45,46]. Here, to extract further information on the respective coral species stress response, we additionally extracted the breakpoint temperature ED5 (5% effective dose), the temperature limit ED95 (95% effective dose), and the decline width (*DW* = ED95 - ED5) for all species and seasons in both heat and cold CBASS assays[76,121]. The *DW* is used as a metric to describe the respective response curve (i.e., $F_v/F_m$ decline with increasing/decreasing temperature), representing how gradually or sharply the photosynthetic performance declines under temperature stress. In this context, a narrow *DW* suggests a steeper decline with a sudden loss of performance once a particular threshold is exceeded, whereas a wide *DW* represents a more gradual decline, indicating greater flexibility or resilience to incremental stress[75,76]. Together, these metrics (i.e., ED50, ED5, ED95, and resulting *DW*) can be used to describe the temperature tolerance of coral species, and the variability and plasticity of the stress response[45,46,76]. Calculating *DW*s can help to distinguish between species that are more sensitive to small temperature changes versus those that can buffer temperature stress over a wider range[75,76].

Photosynthetic efficiencies on all fragments were measured following a 1 h dark-acclimation at the start of the recovery phase and following an 11 h overnight recovery period at the end of the experiment. Each fragment was measured once to avoid PAM saturating light pulses to potentially induce light artifacts. All measurements were taken on the side and middle of the fragments to avoid measurements of the tips or base. The following settings were chosen for the PAM: measuring light intensity of 6, measuring light frequency of 3, Signal Gain of 2, Signal Damping of 4, Electron-Transfer-Rate-Factor (ETR-Factor) of 0.85, and Actinic Light Factor of 1. Both heat and cold CBASS assays were performed using fragments of the same genotype with a biological replication of $n = 10$ for *Acropora* sp. in both heat and cold CBASS runs, $n = 9$ for heat and $n = 8$ for cold CBASS runs with *P. favosa*, and $n = 7$ for heat and $n = 6$ for cold CBASS runs with *S. pistillata*, respectively. Note that different *S. pistillata* colonies were chosen in February 2024 due to mortality after a bleaching event in summer 2023[116,117]. Cold-induced bleaching has been reported for all three species (see Supplementary Data 1), and these species have also been successfully used in previous classical CBASS assays[46].

### ED estimation and statistical analyses

Temperature tolerance thresholds were determined for each species and treatment (cold and heat) as the mean (across all genotypes) temperature at which photosynthetic efficiency dropped to 50% (ED50)[69] using the dose response curve (DRC) package in R[122]. We used the reached temperatures of each assay to calculate the models (see Supplementary Fig. 3). Additionally, cold and heat ED5 and ED95 were also calculated from the DRC curve fitting[76]. Statistical differences among species ED50s and *DW* at each treatment (heat and cold CBASS, respectively) were assessed using a Kruskal-Wallis non-parametric test, followed by Dunn's post hoc test (Holm correction) for pairwise comparisons. Normality was met by all the experiments except the cold CBASS data collected in February 2024, based on the Shapiro-Wilk normality test. Therefore, we decided to run a non-parametric test for all experiments. All ED5, ED50 and ED95 are presented as mean ± standard error of mean.

Photosynthetic efficiency of the species on each of the sampling times (summer and winter) during the CBASS assay was quantified and compared after the stress phase and at the end of the experiment. For this and as both measurements ($t_7$, $t_{18}$) were taken from the same coral fragment, we treated observations as paired and tested the difference across time points ($t_7$, $t_{18}$) using a paired t-test. If the distribution of paired differences was non-normal (Shapiro-Wilk $p < 0.05$) or sample size was small, we used the Wilcoxon signed-rank test instead. Results including mean ± SD at each time point and season and mean difference between post stress ($t_7$) and post recovery ($t_{18}$) are reported in Supplementary Fig. 7 and Table 5. Two-sided $p$-values

were Benjamini-Hochberg adjusted across all coral species x temperatures tests to control for false discovery rates with significance levels at α = 0.05. Further, Spearman rank correlation was performed to identify linkages between calculated heat and cold ED50s and the three most dominant bacterial families (*Endozoicomonadaceae, Rhodobacteraceae, Simkaniaceae*; results see Supplementary Table 1).

### DNA extraction, library construction, and 16S rRNA gene amplicon sequencing

Coral fragments used for microbial community assessment were sampled from the same coral colonies as used for the CBASS assays, following the guidelines described by (Voolstra et al., 2021a). At each sampling time (summer 2023 or winter 2024), one entire fragment for each colony (n = 10 and 9 for *Acropora* sp., *n* = 9 and 8 for *P. favosa*, *n* = 8 and 6 for *S. pistillata*, respectively) was collected and transported to the research vessel in sterile zip bags (Whirl-Pak ®, Nasko Sampling LCC, Chicago, United States). Lower replication in winter resulted from dead coral colonies as a consequence from the summer 2023 bleaching event observed in the Central Red Sea[116,117]. On board, all fragments were immediately snap-frozen in liquid nitrogen, and subsequently placed at −80 °C until further processing. Approximately 0.3 g of each fragment was incubated overnight with 700 μl of DESS buffer (20% dimethyl sulfoxide, 0.25 M ethylenediaminetetraacetic acid, and saturated sodium chloride (NaCl), with adjusted pH 8.0) before DNA extraction. Total DNA was extracted using the DNeasy® Blood and Tissue kit (Qiagen, Hilden, Germany) following the manufacturer's instructions with a Gram-positive pre-treatment step, consisting of a 37 °C incubation for 30 minutes in enzymatic lysis buffer containing lysozyme (20 mg/mL), and a 56 °C 3–4 h incubation in the kit lysis solution and proteinase K at 750 rpm in a Thermomixer (ThermoFisher®). For quantification of the extracted DNA, a high sensitivity Qubit™ dsDNA assay kit and a Qubit 2.0 Fluorometer (Invitrogen™) was used, and then the V3 and V4 regions of the 16S rRNA gene were amplified in triplicate PCRs using the universal primers 341 F 5' CCTACGGGNGGC WGCAG 3' and 785 R 5' GAC TAC HVG GGT ATC TAA TCC 3'. PCR reactions consisted of 1 μl from primer F and R, 8.00 μl of PCR grade water, 12.5 μl of Kapa HiFi HotStart Master Mix (Roche®) and 2.50 μl of sample DNA standardized to a concentration of 5 ng/μl. Thermal cycling conditions were 95 °C for 3 min, followed by 35 cycles 95 °C, 62 °C, and 72° for 30 s each, with a final 5 min extension at 72 °C. Negative reagent controls without template DNA were run for each PCR replicate. Then, PCR products were verified and quantified by mixing their equal volume with 1X loading buffer and performing electrophoresis on 2% agarose gels. Subsequently, PCR plates were stored at -20 °C until library preparation. PCR products were purified using a Qiagen Gel Extraction Kit (Qiagen, Hilden, Germany). Sequencing libraries were generated with a NEBNext® Ultra™ II DNA Library Prep Kit. The library quality was evaluated on a Qubit 2.0 Fluorometer (ThermoScientific™) and Agilent Bioanalyzer 2100 system. Libraries were sequenced on a NovaSeq platform (Illumina), and 250 bp paired-end reads were generated.

### Sequencing data processing workflow and statistical analyses
The 16S rRNA gene amplicon libraries were processed using the DADA2 pipeline[123,124]. Briefly, the raw reads were decontaminated by phiX and adapter-trimmed using the "BBDuk" tool from the BBMap suite (Bushnell B., http://sourceforge.net/projects/bbmap/). Afterwards, PCR primers were removed from the reads using the "cutadapt" tool[125] and the maxEE (maximum expected error) parameter at the filterAndTrim step of DADA2 was set to 6 for the forward and reverse reads. The sequences were analysed under the pseudo-pooling mode by following the standard DADA2 (v1.22) workflow, after performing concatenation of the forward and reverse reads via the "justConcatenate" option in the mergePairs function of DADA2.

All analyses were performed in R v4.3.2 with the R.studio interface (v2022.07.01) using the functions in the 'Phyloseq' package (v1.42.0)[126]. Taxonomy was assigned to ASVs at 99% sequence identity using the SILVA-v138.1 classifier[127]. We removed reads identified as contaminants,

mitochondria, chloroplasts, archaea, eukaryotes, and singletons. Then, the ASV table was rarefied to 100,000 reads (read depth determined by a rarefaction curve; Supplementary Fig. 4) using the 'rarecurve' function for accounting for sequencing coverage and the 'rarefy_even_depth' function in the 'Vegan' package[128]. Alpha diversity was calculated using the 'estimate_richness' function in Phyloseq, with the default diversity indices (observed number of ASVs, Shannon H', and Chao1). For beta diversity calculations, ASVs were transformed to relative abundance to calculate the Bray-Curtis distance matrices that were used as metrics. Moreover, we used the 'betadisper' command in the Vegan package to check for differences in homogeneity of variances[129]. Patterns in beta diversity and permutational multivariate analyses of variance (PERMANOVA) were visualized using principal coordinate analyses (PCoAs) based on Bray Curtis dissimilarity to test for differences in the corals' prokaryotic communities across species and seasons. To identify differentially abundant ASVs across species and seasons, the 'Analysis of the Composition of Microbiomes with Bias Correction' (ANCOM-BC2)[130] was used. This method features certain characteristics: i) it estimates unknown sampling fractions, ii) corrects bias from sample differences, iii) models absolute abundance with linear regression, and iv) provides a statistically valid test with appropriate *p* values, false discovery rate (FDR) control, and sustained power. This analysis was performed on total ASV counts (after removing singletons), using the Benjamin-Hochberg (BH) method to correct for false positives and an alpha = 0.05 for significance. An ASV was considered significant when it was either enriched or decreased significantly (*p-adj.* < 0.05), respectively in a particular species comparing seasons, or vice versa.

### Reporting summary
Further information on research design is available in the Nature Portfolio Reporting Summary linked to this article.

### Data availability
All raw sequence reads were deposited in the ENA under the study accession number PRJEB88613.

### Code availability
The R scripts, raw data, results of differentially abundant ASVs for each species at the different seasons as well as the results of the ANCOM-BC2 analyses are available online[131].

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

## Acknowledgements

The authors thank Zenon Batang and Nabeel M. Alikunhi for allocation of workspace and technical support at the Core Lab for Coastal and Marine Resources (CMOR) at KAUST. This work was supported by KAUST grant number BAS/1/1095-01-01 and KAUST Competitive Research Grant (CRG) URF/1/4723-01-01. Matilde Marques acknowledges a doctoral grant (doi.org/10.54499/SFRH/BD/151376/2021) from the MIT Portugal program, and Tina Keller-Costa a research scientist contract (CEECIND/00788/2017), both financed through the Portuguese Foundation for Science and Technology (FCT).

## Author contributions

Yusuf C. El-Khaled: Conceptualization, Data curation, Formal Analyses, Investigation, Methodology, Visualization, Writing—Original Draft, Writing—Review and Editing; Francisca C. García: Conceptualization, Formal Analysis, Methodology, Writing—Review and Editing; Neus Garcias-Bonet: Conceptualization, Formal Analysis, Investigation, Methodology, Writing—Review and Editing; Matteo Monti: Formal Analysis, Visualization, Writing—Review and Editing; Erika P. Santoro: Investigation, Writing—Review and Editing; Matilde Marques: Investigation, Writing—Review and Editing; Natalie Dunn: Investigation, Writing—Review and Editing; Tina Keller-Costa: Resources, Writing—Review and Editing; Christian R. Voolstra: Conceptualization, Formal Analysis, Methodology, Writing—Review and Editing; Raquel S. Peixoto: Conceptualization, Funding acquisition, Methodology, Resources, Writing—Review and Editing.

## Competing interests

The authors declare no competing interests.
