## [Transparent Peer Review file · Communications Biology]

Resolving Coral Temperature Vulnerability through Heat and Cold Bleaching Thresholds

Corresponding Author: Dr Yusuf El-Khaled

Version 0:

Reviewer comments:

Reviewer #1

(Remarks to the Author)

Review:

El-Khaled et al. Resolving Coral Temperature Vulnerability through Heat and Cold Bleaching Thresholds

Generally, I found the paper well-constructed and I recommend publication. I have a very few comments, listed below:

Methods:

Lines 509-510: So the acclimation time between collection and CBASS initiation was 3 hrs? How different was temperature at the wet lab from that of the CPV? What was your basis for using such a short acclimation period?

Lines 521-522: were the maximum and minimum temperatures in January the same? 24oC? Or is there a number missing here?

Lines 522-525: this is confusing; could you please just state the temperature profile that you ended up using, just for clarity?

Lines 526-529: Can you clarify whether you used separate (but clonal?) frags for heat and cold assays or used the same fragments subjected to both assays? It's a little unclear.

Discussion:

Lines 266-267: do you mean "both hot and cold CBASS...?"

Lines 278-280: You may want to include here some discussion of the link between bleaching severity in the summer and warm winter water—modelling work done based on GBR corals. It would be relevant here.

Reviewer #2

(Remarks to the Author)

The manuscript by El-Khaled et al. investigates the effects of acute heat and cold stress on three species of scleractinian corals from the Red Sea, utilizing an automated CBASS system. The authors assess changes in photosynthetic efficiency and temporal shifts in the coral-associated microbiome under different temperature treatments to analyze the thermal tolerance ranges of the studied species. It is particularly valuable to see a study that not only examines heat tolerance profiles but also brings attention to the understudied impacts of cold stress on coral resilience.

Introduction:

The introduction is thorough and provides a valuable overview of current knowledge on cold stress in corals. However, I would appreciate a bit more context regarding the experimental framework. While this information does not necessarily need to be included in the Introduction, it would strengthen the manuscript if the authors briefly explained what PAM measurements represent—specifically, what they serve as a proxy for, their biological significance, how they relate to coral bleaching, and any known limitations (e.g., that changes in photosynthetic efficiency may not directly reflect disruptions to the coral-algal symbiosis typically associated with bleaching). Similarly, it would be helpful to include a brief explanation of additional metrics such as ED5, ED95, and decline width, and to clarify their biological relevance in the context of thermal tolerance.

Methods:

The first paragraph of the methods section seems to focus more on contextual information and the rationale behind the experimental layout rather than describing specific methodological steps. As such, it may be more appropriate to move it to

the introduction or the beginning of the results section.

I was confused by the number of coral fragments used for the study. Was every coral colony represented by only one fragment per treatment? And then there were two additional fragments collected from each colony, one in winter and one in the summer, that were tested for microbiome structure? So there were no technical replicates for the temperature treatments? If so, how did the authors account for tank effects, especially if different tanks were used across treatments, which can interfere with biological responses?

L521-522 "...we similarly chose the mean maximum and minimum temperature of January 2024 (24 C)..." The sentence suggests the authors chose two temperatures, while they indicate only one temperature.

ED Estimation. I have a different view on how to calculate ED50 in this particular experimental setting. I believe that plotting Fv/Fm over raw temperature is informative in experiments where the same fragment is repeatedly measured as temperature increases incrementally, since, in that case, the difference in Fv/Fm could be directly attributed to a specific temperature change. However, in this study, each temperature treatment involved a separate set of fragments exposed to distinct thermal profiles. As a result, the variation among treatments likely reflects not just the maximum temperature but also differences in accumulated heat exposure. For this, I believe that using metrics similar to the widely used degree heating week, which represents the entirety of accumulated heat stress relative to the control treatment, would be a better fit.

In the same manner, when looking at the temperature profiles in Supplementary Figure S3, the authors show t_0 as a pre-set temperature. But it is unclear what happened to corals between the time they were collected from the sea until they reached the given temperature. This is particularly relevant in the winter treatment, where corals likely had to be warmed up to reach the t_0 temperature, potentially introducing an initial heat stress event. If this pre-exposure heating did not occur in the summer treatment, then comparing thermal stress responses between summer and winter treatments using raw temperature metrics may be problematic, as the corals in each setting would have experienced notably different thermal trajectories.

L196 – I wasn't able to find any graphical representation of correlations, nor any supplementary data detailing the correlation analysis. I believe a dot plot or similar, that would show the relationship, would be very helpful.

L203 – 210 I think this paragraph shouldn't be part of the bacterial profiling

Recovery analysis: I found the recovery analysis somewhat difficult to follow, in part because not all relevant data appear to be presented or fully discussed in the manuscript. Also, what was used as a control? Could we see how Fv/Fm values were changing in control treatments (represented by yellow and light blue lines) compared to all other temperature profiles? Regarding Figures 5 and S7, it is not entirely clear why the authors chose to display data from only selected temperatures. Would it be possible to show recovery data for all temperature profiles and all coral species, either in the main text or in the supplementary materials? This would greatly improve transparency and help the reader better understand species- and treatment-specific recovery patterns.

Discussion

I appreciate that the authors have structured the Discussion section by topic, which enhances clarity and readability. However, at times it was difficult to clearly distinguish between the findings of this study and those of previously published work (for example, in lines 304–306 and 316–320). To improve clarity, I suggest that the authors explicitly indicate which results stem from their own data and which are drawn from the existing literature. This distinction would help readers better appreciate the novel contributions of the current study.

L337-338. While I agree with this statement, I would maybe add that (at least in my view), this work contributes to the growing body of evidence that nowadays, corals live much closer to their upper thermal limit than mid-range or lower thermal limits, which makes them that much more vulnerable to temperature extremes.

I am not a microbiologist, so I have a naïve but genuine question: is it known whether the bacterial taxa discussed in this study also have defined upper and lower thermal limits, similar to corals? Could some of the observed seasonal shifts in microbial communities be explained, at least in part, by temperature-dependent differences in bacterial growth rates—for example, certain taxa proliferating more slowly in winter conditions but more rapidly in summer? If so, it might be helpful for the authors to briefly address whether this kind of temperature-driven microbial dynamics could contribute to the observed patterns, in addition to host-related or environmental factors.

L403-410 sounds more like results than discussion to me.

L431 -433 I didn't understand the explanation of what these two different timepoints are used as a proxy for. Could the authors rephrase it, please?

L436 and L445 both point to Figure 5, but describe different results in my understanding.

L448 "... all coral species appear to possess mechanisms to cope with extremely low temperatures, expressed via an increased photosynthetic efficiency..." This makes it sound like corals in cold stress increase their photosynthetic efficiency. What figure shows that? Where is the comparison to a control treatment and a relevant statistical analysis supporting this claim?

Reviewer #3

(Remarks to the Author)

The study, "Resolving coral temperature vulnerability through heat and cold bleaching thresholds" by El-Khaled et al, aims to examine and determine thermal thresholds for corals under both elevated temperature and cold bleaching conditions using Red Sea corals. The authors use the well-described short-term stress system CBASS to determine bleaching thresholds for three coral species. While the elevated temperature bleaching thresholds for these corals have already been worked out, the novelty of this study is that the same standardized system is employed to test the thresholds of bleaching in response to cold temperatures. We know that corals can bleach and indeed die under cold stress, but as the authors point out, there is yet to be a controlled test of these cold thresholds in such corals.

The other novel aspect of the study, of course, is the additional microbiome analyses that accompany the bleaching threshold data. Showing changes in the microbiome in response to heat or cold stress and that not all of the corals responded in the same way provides some interesting commentary to the discussion of how microbes may contribute to the thermal resilience of their coral hosts.

This study is interesting and demonstrates an important use for the CBASS system, now available for hot-CBASS and cold-CBASS studies, but as the authors do indicate, is it likely that corals will be seeing these 'cold stress' conditions as global climate change continues to warm the tropical waters where corals are found? Technically, this study is interesting, but I am not sure if the ability to determine cold bleaching thresholds is mission critical for the conservation and protection of corals into the warmer futures that they will experience.

The writing is clear and well-explained (some minor comments below). I did find the Discussion a bit verbose and I believe the text could be shortened.

Overall, this is a valuable study that was well thought out and performed. The statistics are appropriate and it is clear that the research team was well equipped to conduct this study and analyses.

Some minor comments:

Line 32: there is an extra comma after sp.

Line 55: comma not needed after "temperature"

Line 81: extra space before reference citation

Line 132: The presentation of the bleaching thresholds to such a fine decimal is tricky since you only tested stress at 3 degree intervals. Perhaps this precision has already been demonstrated using the CBASS system, but that is some real fine scale precision and I wonder if it might be more fair to round those numbers a bit.

Line 224: Figure 3, it is very challenging to distinguish between the various shades of colors on this bacterial abundance plot. Please consider changing the color ranges.

Line 336: Is this 6-9°C temperature range shown here linked to the natural temperature range in the environment by season for these corals?

Line 359: What about the microbiome of the surrounding seawater? If some of these coral species are known to have microbiomes influenced by the environment, could the changes noted here be due to changes in the microbial composition in seawater? I don't remember seeing anything about sampling of the water around the corals.

Line 428: change to "Here, we"

Line 447: Did you see any direct damage to the chloroplasts as a result of the cold stress as seen in studies like Bieri et al 2016? They did use 4°C cold shock but the cold has been reported to cause severe damage to the photosystem.

Line 522: Perhaps clarify that the stressed temperatures were 24, 21, 18, 15°C?

Version 1:

Reviewer comments:

Reviewer #2

(Remarks to the Author)

I thank the authors for their careful and thorough revision of the manuscript. They have addressed the comments of all reviewers in a constructive way, and the discussion throughout the process has been very productive. I recommend that the manuscript be accepted for publication in its current form.

Reviewer #3

(Remarks to the Author)

I thank the authors for addressing my comments and concerns.

#	Paragraph/ Lines	Comments Reviewer 1	Author's action	Author's response
1	General	Generally, I found the paper well-constructed and I recommend publication. I have a very few comments, listed below:	Appreciated.	Thank you for the valuable suggestions and feedback!
2	509-510	So the acclimation time between collection and CBASS initiation was 3 hrs? How different was temperature at the wet lab from that of the CPV? What was your basis for using such a short acclimation period?	Changed	We used in situ temperatures during this 3h period. CBASS assays are designed to be performed as close to in situ conditions as possible, i.e., bypassing any possible acclimation to avoid adjustments to a possible tank condition (see Voolstra et al. 2020, Global Change Biology). Edits: 1.571-573: “. During these three hours, coral fragments were kept under ambient temperatures aiming to bypass any possible acclimation and to avoid adjustments to possible tank conditions aiming to have close to in-situ conditions⁴⁵.”
3	521-522	were the maximum and minimum temperatures in January the same? 24oC? Or is there a number missing here?	Changed.	The mean minimum temperatures were 24°C and were chosen as a starting point for both cold and heat CBASS assays. However, the chosen temperatures for the heat CBASS assay (24, 27, 30 and 33°C) did not lead to a reduction of photosynthetic efficiency (as shown in Supplementary Figure S. 1). We therefore used the same target temperature profile for the heat CBASS as in summer, i.e., 31, 34, 37 and 40°C (see Supplementary Fig. S3). We have added additional information to the manuscript. Edits: 1.568-574: “In February, we similarly choose the mean minimum temperature of January 2024 (24°C, respectively; Supplementary Fig. S2) to apply the heat (24, 27, 30, 33°C) and cold (24, 21, 18, 15°C) stress temperature profiles. However, the initially chosen temperatures for the heat CBASS assay (24, 27, 30, 33°C) did not alter the photosynthetic efficiency that allowed the calculation of bleaching thresholds (see results in

				Supplementary Fig. S1). We therefore choose the same target temperature profile as performed before in summer 2023, i.e., 31, 34, 37 and 40°C (Supplementary Fig. S3).”
4	522-525	this is confusing; could you please just state the temperature profile that you ended up using, just for clarity?	Changed.	We agree with the reviewer. We therefore added the respective temperature information. In winter, the mean minimum temperatures were 24°C and were chosen as a starting point for both cold and heat CBASS assays. However, the chosen temperatures for the heat CBASS assay (24, 27, 30 and 33°C) did not lead to a reduction of photosynthetic efficiency. We therefore used the same temperature profile for the heat CBASS as in summer (i.e., 31, 34, 37 and 40°C). We have added additional information to the manuscript. Edits: 1.568-575: “In February, we similarly choose the mean minimum temperature of January 2024 (24°C, respectively; Supplementary Fig. S2) to apply the heat (24, 27, 30, 33°C) and cold (24, 21, 18, 15°C) stress temperature profiles. However, the initially chosen temperatures for the heat CBASS assay (24, 27, 30, 33°C) did not alter the photosynthetic efficiency that allowed the calculation of bleaching thresholds (see results in Supplementary Fig. S1). We therefore choose the same target temperature profile as performed before in summer 2023, i.e., 31, 34, 37 and 40°C (Supplementary Fig. S3). In-situ temperatures for July 2023 and January 2024 from the same location are displayed in Supplementary Fig. S2.”
5	526-529	Can you clarify whether you used separate (but clonal?) frags for heat and cold assays or used the same fragments subjected to both assays? It’s a little unclear.	Changed.	We used distinct but clonal fragments for each assay in both summer and winter. We have clarified this in the main manuscript. Edits: 1. 575-579: “Coral fragments for both heat and cold CBASS assays were collected from the same set of tagged coral colonies (i.e., distinct but clonal fragments for each assay). For cold CBASS assays, coral fragments were collected on August 15th,

				2023, and on February 14th, 2024; coral fragments for heat CBASS were collected on August 20 th , 2023, and on February 18 th , 2024, respectively.“
6	266-267	do you mean “both hot and cold CBASS...”?	Changed.	Yes, we were referring to both hot and cold CBASS. We clarified this in the main manuscript. Edits: 1.277-282: “While the classical CBASS temperature profiles are typically chosen based on local maximum monthly mean seawater temperatures of a given site or an average temperature of multiple sites⁴⁶, we here adapted the profiles for both hot and cold CBASS according to the respective season (i.e., maximum mean summer and winter, respectively) to achieve a decline in photosynthetic efficiency that allowed the calculation of corresponding ED50 values.”
7	278-280	You may want to include here some discussion of the link between bleaching severity in the summer and warm winter water—modelling work done based on GBR corals. It would be relevant here.	Changed.	We agree that the reviewer’s suggestion provides further depth to the discussion. We reference studies by Cornet et al. (2025, Biological Conservation) and Mellin et al. 2024 (Science Advances) to strengthen our discussion. Edits: 1.291-300: “Additionally, consistent with this seasonal framing, modelling studies from the Great Barrier Reef show that anomalously warm, “hot winters”, can precondition reefs and are associated with greater bleaching severity in the subsequent summer, for example during the 2016-2017 years⁷². Accordingly, prediction frameworks should explicitly incorporate winter-heat and prior-year heat metrics, not only summer exposure, because winter conditions help structure summer bleaching risk^{72,73}. Therefore, we propose that determining the temperature tolerance range using both cold and heat CBASS assays as a critical tool complementing current CBASS assays allowing the determination of viable temperature ranges that comprise the entire spectrum, i.e., cold and hot

				temperature tolerance thresholds, of the coral species under investigation.“
--	--	--	--	--

#	Paragraph/ Lines	Comments Reviewer 2	Author's action	Author's response
1	General	The manuscript by El-Khaled et al. investigates the effects of acute heat and cold stress on three species of scleractinian corals from the Red Sea, utilizing an automated CBASS system. The authors assess changes in photosynthetic efficiency and temporal shifts in the coral-associated microbiome under different temperature treatments to analyze the thermal tolerance ranges of the studied species. It is particularly valuable to see a study that not only examines heat tolerance profiles but also brings attention to the understudied impacts of cold stress on coral resilience.	Appreciated.	We are thankful for the very thorough review and feedback provided by the reviewer.
2	Introduction	The introduction is thorough and provides a valuable overview of current knowledge on cold stress in corals. However, I would appreciate a bit more context regarding the experimental framework. While this information does not necessarily need to be included in the Introduction, it would strengthen the manuscript if the authors briefly explained what PAM measurements represent—specifically, what they serve as a proxy for, their biological significance, how they relate to coral bleaching, and any known limitations (e.g., that changes in photosynthetic efficiency may not directly reflect disruptions to the coral-algal symbiosis typically associated with bleaching). Similarly, it would be helpful to include a brief explanation of additional metrics such as ED5, ED95, and decline width, and to clarify their biological relevance in the context of thermal tolerance.	Changed.	We agree with the points raised by the reviewer and have added respective information to both the Introduction and the Method section of the manuscript. We used Warner et al. (1999), Schreiber et al. (1995), Ralph & Gademann (2005), Voolstra et al. (2020, 2025), Evensen et al. (2023) and Gomez-Campo & Baums (2024) to describe the metrics used in the present study. Edits: 1.109-111: “By introducing a novel cold effective dose (ED) 50, i.e., an empirically derived threshold when a decrease of 50% relative to the control is reached, and directly comparing it with heat ED50 values, we establish species-specific viable temperature ranges.” 1.598-615: “Photosynthetic efficiency serves as a non-invasive indicator of algal photo-physiology¹¹⁸, and reflects the performance of the symbionts’ photosystem¹¹⁹. A decline in F_v/F_m usually indicates photodamage, impaired electron transport, or protective downregulation in response to stress. Declines in F_v/F_m often precede visible bleaching, with

				associated ED50 thresholds being used as thermal tolerance proxies informing bleaching susceptibility^{45,46}. Here, to extract further information on the respective coral species stress response, we additionally extracted the breakpoint temperature ED5 (5% effective dose), the thermal limit ED95 (95% effective dose), and the decline width ($DW = ED95 - ED5$) for all species and seasons in both heat and cold CBASS assays^{76,120}. The DW is used as a metric to describe the respective response curves (i.e., F_v/F_m decline with increasing/decreasing temperature), representing how gradually or sharply the photosynthetic performance declines under temperature stress. In this context, a narrow DW suggests a steeper decline with a sudden loss of performance once a particular threshold is exceeded, whereas a wide DW represents a more gradual decline, indicating greater flexibility or resilience to incremental stress^{75,76}. Together, these metrics (i.e., ED50, ED5, ED95, and resulting DW) can be used to describe the temperature tolerance of coral species, and the variability and plasticity of the stress response^{45,46,76}. Calculating DWs can help to distinguish between species that are more sensitive to small temperature changes versus those that can buffer temperature stress over a wider range^{75,76}.”
3	Methods	The first paragraph of the methods section seems to focus more on contextual information and the rationale behind the experimental layout rather than describing specific methodological steps. As such, it may be more appropriate to move it to the introduction or the beginning of the results section.	Changed.	We acknowledge the reviewer’s point and therefore moved the cold bleaching observations during winters of 2023 and 2025 to the introduction section to provide a better rationale. We, however, suggest retaining the study site description and the quantification of background in situ parameters in the Methods section. Edits: 1.92-99: “During the winters of 2023 and 2025, we observed pale and bleached P. verrucosa, Acropora sp., and Echinopora sp. coral colonies (Fig.1, Supplementary Table S1) when in-situ temperatures were lowest in the Central Red Sea. Additionally, bleached Galaxea fascicularis colonies were also observed in-situ in winter 2023 during a long-term monitoring study (Dunn

				et al., unpublished data). Despite a considerable number of reports on cold-bleaching showing that cold bleaching can occur globally affecting a considerable number of coral species (Fig. 1, Supplementary Table S1), empirical data remain sparse and fragmented, especially when compared to the wealth of research on heat-induced bleaching. “
4	Methods	I was confused by the number of coral fragments used for the study. Was every coral colony represented by only one fragment per treatment? And then there were two additional fragments collected from each colony, one in winter and one in the summer, that were tested for microbiome structure? So there were no technical replicates for the temperature treatments? If so, how did the authors account for tank effects, especially if different tanks were used across treatments, which can interfere with biological responses?	Noted and clarified.	We thank the reviewer for raising this important point regarding the experimental design. In our study, each coral colony was represented by a single fragment per treatment temperature in the CBASS assays, and two additional fragments per colony were collected for microbiome characterization (one in summer and one in winter). We deliberately minimized the number of fragments per colony to reduce ecological impact, while opting to maximize the number of colony (biological) replicates that entail a component of technical variance. This approach inherently captures technical variance and follows established CBASS methodology (e.g., Voolstra et al. 2020, Global Change Biology; Evensen et al. 2021, Limnology & Oceanography; Alderdice et al. 2022, Scientific Reports). Regarding tank effects, previous CBASS studies have shown that the biological signal of coral thermal responses is substantially stronger than potential tank effects, making colony-level replication the critical factor for deriving robust bleaching thresholds. Edits: 1.575-579: “Coral fragments for both heat and cold CBASS assays were collected from the same set of tagged coral colonies (i.e., distinct but clonal fragments for each assay). For cold CBASS assays, coral fragments were collected on August 15th, 2023 and on February 14th, coral fragments for heat CBASS were collected on August 20th, 2023, and on February 18th, 2024, respectively.”

5	521-522	“...we similarly chose the mean maximum and minimum temperature of January 2024 (24 C)...” The sentence suggests the authors chose two temperatures, while they indicate only one temperature.	Changed.	We agree that this is misleading. We added the respective temperature information to the manuscript. In winter, the mean minimum temperatures were 24°C and were chosen as a starting point for both cold and heat CBASS assays. However, the chosen temperatures for the heat CBASS assay (24, 27, 30 and 33°C) did not lead to a reduction of photosynthetic efficiency. We therefore used the same temperature profile for the heat CBASS as in summer (i.e., 31, 34, 37 and 40°C). We have added additional information to the manuscript. Edits: 1.568-575: “In February, we similarly choose the mean minimum temperature of January 2024 (24°C, respectively; Supplementary Fig. S2) to apply the heat (24, 27, 30, 33°C) and cold (24, 21, 18, 15°C) stress temperature profiles. However, the initially chosen temperatures for the heat CBASS assay (24, 27, 30, 33°C) did not alter the photosynthetic efficiency that allowed the calculation of bleaching thresholds (see results in Supplementary Fig. S1). We therefore choose the same temperature profile as performed before in summer 2023, i.e., 31, 34, 37 and 40°C. In-situ temperatures for July 2023 and January 2024 from the same location are displayed in Supplementary Fig. S2.”
6	ED Estimation	I have a different view on how to calculate ED50 in this particular experimental setting. I believe that plotting Fv/Fm over raw temperature is informative in experiments where the same fragment is repeatedly measured as temperature increases incrementally, since, in that case, the difference in Fv/Fm could be directly attributed to a specific temperature change. However, in this study, each temperature treatment involved a separate set of fragments exposed to distinct thermal profiles. As a result, the variation among treatments likely reflects not just the maximum	Clarified.	We thank the reviewer for raising this important point. We agree that cumulative exposure would be a concern if the same fragments were exposed sequentially to increasing temperatures. However, in the CBASS design each colony was represented by clonal fragments exposed independently to the full temperature range, and ED50 values were modeled separately for heat and cold CBASS assays. This ensures that the derived thresholds reflect acute responses to distinct thermal profiles rather than cumulative heat exposure. Our approach follows the established CBASS framework, which is designed specifically to capture instantaneous thermal tolerance limits (Voolstra et al. 2020, Global Change Biology).

		temperature but also differences in accumulated heat exposure. For this, I believe that using metrics similar to the widely used degree heating week, which represents the entirety of accumulated heat stress relative to the control treatment, would be a better fit.		
7	ED Estimation	In the same manner, when looking at the temperature profiles in Supplementary Figure S3, the authors show t0 as a pre-set temperature. But it is unclear what happened to corals between the time they were collected from the sea until they reached the given temperature. This is particularly relevant in the winter treatment, where corals likely had to be warmed up to reach the t0 temperature, potentially introducing an initial heat stress event. If this pre-exposure heating did not occur in the summer treatment, then comparing thermal stress responses between summer and winter treatments using raw temperature metrics may be problematic, as the corals in each setting would have experienced notably different thermal trajectories.	Noted.	We thank the reviewer for this observation. We clarify that CBASS is designed to empirically capture the full thermal tolerance range of corals by exposing clonal fragments to discrete temperature profiles. In winter, we initially tested temperatures based on in situ conditions (24, 27, 30, 33 °C), but these did not elicit measurable ED50 thresholds. We therefore repeated the assay with the same profile used in summer (31, 34, 37, 40 °C) to ensure comparability across seasons and to capture the upper tolerance limits. This approach reflects the standard CBASS framework, which assesses acute thermal thresholds independent of the precise in situ starting temperature. The resulting data highlight that some coral species show limited seasonal differences in thermal tolerance, consistent with previous findings that not all corals display strong seasonal acclimatization. Edits: 1.568-574: “In February, we similarly choose the mean minimum temperature of January 2024 (24°C, respectively; Supplementary Fig. S2) to apply the heat (24, 27, 30, 33°C) and cold (24, 21, 18, 15°C) stress temperature profiles. However, the initially chosen temperatures for the heat CBASS assay (24, 27, 30, 33°C) did not alter the photosynthetic efficiency that allowed the calculation of bleaching thresholds (see results in Supplementary Fig. S1). We therefore choose the same target temperature profile as performed before in summer 2023, i.e., 31, 34, 37 and 40°C (Supplementary Fig. S3).“

8	196	I wasn't able to find any graphical representation of correlations, nor any supplementary data detailing the correlation analysis. I believe a dot plot or similar, that would show the relationship, would be very helpful.	Edited.	We added information that details of the correlation analysis that can be found in Supplementary Table S2. Also, details of the calculation are included in the methods section. Edits: 1.646-658: "For this and as both measurements (t_7, t_{18}) were taken from the same coral fragment, we treated observations as paired and tested the difference across time points (t_7, t_{18}) using a paired t-test. If the distribution of paired differences was non-normal (Shapiro-Wilk $p < 0.05$) or sample size was small, we used the Wilcoxon signed-rank test instead. Results including mean \pm SD at each time point and season and mean difference between post stress (t_7) and post recovery (t_{18}) are reported in Supplementary Figure S7 and Table S5. Two-sided p-values were Benjamini-Hochberg adjusted across all coral species x temperatures tests to control for false discovery rates with significance levels at $\alpha = 0.05$. Further, Spearman rank correlation was performed to identify linkages between calculated heat and cold ED50s and the three most dominant bacterial families (Endozoicomonadaceae, Rhodobacteraceae, Simkaniaceae; results see Supplementary Table S2)."
9	203 – 210	I think this paragraph shouldn't be part of the bacterial profiling	Edited.	We agree to the reviewer's comment; we therefore added a subheading for the recovery analysis.
10	Recovery analysis	I found the recovery analysis somewhat difficult to follow, in part because not all relevant data appear to be presented or fully discussed in the manuscript. Also, what was used as a control? Could we see how Fv/Fm values were changing in control treatments (represented by yellow and light blue lines) compared to all other temperature profiles?	Edited.	We have added further details (see next comment).
11	Fig 5 and S7	Regarding Figures 5 and S7, it is not entirely clear why the authors chose to display data from only selected temperatures. Would it be possible to show recovery data for all temperature profiles	Edited.	We agree with the point raised by the reviewer. For this, we have re-performed the recovery analysis with a more robust and conservative approach using paired t-tests, or Wilcoxon signed-rank tests with Benjamini Hochberg adjustments. We kept an

	and all coral species, either in the main text or in the supplementary materials? This would greatly improve transparency and help the reader better understand species- and treatment-specific recovery patterns.	updated Figure 5 displaying the recovery capacities after extreme cold stress as we believe that these are the most relevant results for the narrative of the present manuscript but portrayed all data for full transparency in the supplementary (Supplementary Figure S7). We have updated the respective sections in the Methods, Results and Discussion accordingly. Edits: 1.212-223: “Following exposure to extreme cold temperatures in winter (15°C), a significant increase in photosynthetic efficiency was observed across all investigated coral species after the recovery phase (paired t-test $p_{BH-adj} = 0.02$ for Acropora sp., paired t-test $p_{BH-adj} = 0.037$ for P. verrucosa, paired t-test $p_{BH-adj} = 0.007$ for S. pistillata; Fig. 5B; Supplementary Table S5). During summer, no significant recovery of F_v/F_m values was observed for the coldest treatment in any coral species (Fig. 5A), except for a significant increase in F_v/F_m values only recorded for Acropora sp. after mild cold stress at 27°C (paired t-test $p_{BH-adj} = 0.038$; Supplementary Fig. S7, Supplementary Table S5). In contrast, after exposure to extreme heat stress (39–40°C), photosynthetic efficiency remained consistent or decreased across all species and both seasons (Supplementary Fig. S7, Supplementary Table S5).“ 1.463-476:“ Here, we assessed viable temperature ranges by determining the photosynthetic efficiency of three coral species using the classical and ‘cold’ CBASS assays over two seasons. CBASS assays have been designed with photosynthetic efficiency measurements being taken after the stress phase (i.e., t_7), and in some cases, after a subsequent overnight recovery period (i.e., t_{18})⁴⁶. While the t_7 measurements are used to resolve differences in thermal tolerances between coral populations^{45,47,69,75}, measurements at t_{18} allow to assess differences in the response to temperature stress⁹⁰, hypothetically in the form of species-specific recovery
--	---	---

				potentials after temperature stress. Therefore, we here compared the photosynthetic efficiency measured after the stress phase (t_7) and a subsequent 11h recovery phase (t_{18}; Fig 5, Supplementary Fig. S7). Following heat stress in the hot CBASS assays, the photosynthetic efficiency remained similar or decreased for all species and specimens in both summer and winter.“ 1.644-653: “Photosynthetic efficiency of the species on each of the sampling times (summer and winter) during the CBASS assay was quantified and compared after the stress phase and at the end of the experiment. For this and as both measurements (t_7, t_{18}) were taken from the same coral fragment, we treated observations as paired and tested the difference across time points (t_7, t_{18}) using a paired t-test. If the distribution of paired differences was non-normal (Shapiro-Wilk $p < 0.05$) or sample size was small, we used the Wilcoxon signed-rank test instead. Results including mean \pm SD at each time point and season and mean difference between post stress (t_7) and post recovery (t_{18}) are reported in Supplementary Figure S7 and Table S5. Two-sided p-values were Benjamini-Hochberg adjusted across all coral species x temperatures tests to control for false discovery rates with significance levels at $\alpha = 0.05$.”
12	Discussion	I appreciate that the authors have structured the Discussion section by topic, which enhances clarity and readability. However, at times it was difficult to clearly distinguish between the findings of this study and those of previously published work (for example, in lines 304–306 and 316–320). To improve clarity, I suggest that the authors explicitly indicate which results stem from their own data and which are drawn from the existing literature. This distinction would help readers better appreciate the novel contributions of the current study.	Edited.	We agree with the point raised by the reviewer. We have clarified this throughout the discussion. Edits: 1. 317-319: “The distinct response patterns in photosynthetic efficiency during heat assays observed in the present study were less pronounced during winter, with similar DW values observed across species (Supplementary Fig. S5).“ 1.331-335:” Specifically, S. pistillata showed stable F_v/F_m values until reaching a critical threshold, analogous to Acropora sp. under summer heat stress (Fig. 2C, 2B). Conversely, P.

				verrucosa and particularly Acropora sp. exhibited declining photosynthetic efficiency correlated with cold stress intensity during winter, similar to P. verrucosa and S. pistillata responses to summer heat stress observed in the present (Fig. 2B) and a previous study⁷⁸.”
13	337-338	While I agree with this statement, I would maybe add that (at least in my view), this work contributes to the growing body of evidence that nowadays, corals live much closer to their upper thermal limit than mid-range or lower thermal limits, which makes them that much more vulnerable to temperature extremes.	Added.	We agreed to the point raised by the reviewer and added the requested information using Klepac & Barshis (2019; Proc. R. Soc. B) and DeCarlo et al. (2019, Proc. R. Soc. B). Edits: 1.357-361: “This suggests natural acclimatization to naturally fluctuating temperatures with species-specific response patterns to both heat and cold temperature stress. Additionally, these findings complement that tropical reef-building corals live close to their upper thermal tolerance limits, such that even small (~1–3 °C) summer anomalies can trigger bleaching^{80,81}.”
14	Bacterial Profiling	I am not a microbiologist, so I have a naïve but genuine question: is it known whether the bacterial taxa discussed in this study also have defined upper and lower thermal limits, similar to corals? Could some of the observed seasonal shifts in microbial communities be explained, at least in part, by temperature-dependent differences in bacterial growth rates—for example, certain taxa proliferating more slowly in winter conditions but more rapidly in summer? If so, it might be helpful for the authors to briefly address whether this kind of temperature-driven microbial dynamics could contribute to the observed patterns, in addition to host-related or environmental factors.		We thank the reviewer for this thoughtful comment. While our dataset does not directly test temperature-dependent microbial dynamics, we agree that bacterial taxa likely exhibit thermal niches and growth optima that can shift community composition seasonally. Such shifts could contribute to the observed patterns alongside host physiology and environmental factors. We have added a sentence acknowledging that microbial populations may differ in upper/lower thermal limits and competitive performance across seasons. We now highlight this as an important avenue for future work and an alternative mechanism to host-related or environmental factors. Edits: 1.376-378: “Potentially, seasonal restructuring of coral microbiomes may in part reflect temperature-dependent growth optima of bacterial taxa, as seawater temperature is a major driver of microbial community dynamics^{62,85}.”

15	403-410	sounds more like results than discussion to me.	Edited.	We reduced this paragraph to avoid repetition. Edits: 1.433-438: “Although no significant seasonal shifts were detected at the family level in Acropora sp., we identified 69 ASVs that differed between seasons, compared to 49 in P. verrucosa. In contrast, S. pistillata showed no differentially abundant ASVs despite overall seasonal bacteriome changes. Among the ASVs enriched in winter for Acropora sp., the most significant belonged to the family Simkaniaceae (14 ASVs; Fig. 4), a group within Chlamydiia commonly associated with corals.”
16	431-433	I didn’t understand the explanation of what these two different timepoints are used as a proxy for. Could the authors rephrase it, please?	Edited.	We agree to the point raised by the reviewer and have clarified this part of the discussion. Edits: 1.467-472: “While the t_7 measurements are used to resolve differences in thermal tolerances between coral populations^{45,47,69,75}, measurements at t_{18} allow to assess differences in the response to temperature stress⁹⁰, hypothetically in the form of species-specific recovery potentials after temperature stress. Therefore, we here compared the photosynthetic efficiency measured after the stress phase (t_7) and a subsequent 11h recovery phase (t_{18}) of the most extreme temperatures (Fig 5, Supplementary Fig. S7).”
17	436 & 445	both point to Figure 5, but describe different results in my understanding.	Noted.	We have clarified this (see previous comment).
18	448	“... all coral species appear to possess mechanisms to cope with extremely low temperatures, expressed via an increased photosynthetic efficiency...” This makes it sound like corals in cold stress increase their photosynthetic efficiency. What figure shows that? Where is the comparison to a control	Edited.	The recovery analysis and its results are based on a comparison of photosynthetic efficiencies measured directly after the stress phase, and after an 11-hour recovery phase at control temperatures. Information on the statistical analysis is provided in the method section (1. 644-658), and the results are presented in Figure 5, Supplementary Figure S7, and Supplementary Table

	treatment and a relevant statistical analysis supporting this claim?		S5 We also refer to a previous comment that addressed the recovery analysis performed in the present study. Edits: 1.463-469: “CBASS assays have been designed with photosynthetic efficiency measurements being taken after the stress phase (i.e., t_7), and in some cases, after a subsequent overnight recovery period (i.e., t_{18})⁴⁶. While the t_7 measurements are used to resolve differences in thermal tolerances between coral populations^{45,47,69,75}, measurements at t_{18} allow to assess differences in the response to temperature stress⁹¹, hypothetically in the form of species-specific recovery potentials after temperature stress.” 1.481-484: “In contrast to heat stress, we observed a significant increase in the photosynthetic efficiency for all investigated coral species in winter, and for S. pistillata in summer following an 11-h recovery phase at control temperatures after being exposed to extremely cold temperatures (Fig. 5). “
--	--	--	--

#	Paragraph/ Lines	Comments Reviewer 3	Author's action	Author's response
1	General	The study, "Resolving coral temperature vulnerability through heat and cold bleaching thresholds" by El-Khaled et al, aims to examine and determine thermal thresholds for corals under both elevated temperature and cold bleaching conditions using Red Sea corals. The authors use the well-described short-term stress system CBASS to determine bleaching thresholds for three coral species. While the elevated temperature bleaching thresholds for these corals have already been worked out, the novelty of this study is that the same standardized system is employed to test the thresholds of bleaching in response to cold temperatures. We know that corals can bleach and indeed die under cold stress, but as the authors point out, there is yet to be a controlled test of these cold thresholds in such corals. The other novel aspect of the study, of course, is the additional microbiome analyses that accompany the bleaching threshold data. Showing changes in the microbiome in response to heat or cold stress and that not all of the corals responded in the same way provides some interesting commentary to the discussion of how microbes may contribute to the thermal resilience of their coral hosts.	Appreciated.	We are thankful for the positive feedback provided by the reviewer.
2	General	This study is interesting and demonstrates an important use for the CBASS system, now available for hot-CBASS and cold-CBASS studies, but as the authors do indicate, is it likely that corals will be seeing these 'cold stress' conditions as global climate change continues to	Edited.	We appreciate the point raised by the reviewer. The novelty and importance of this study is, besides using cold CBASS assay approaches to evaluate temperature tolerance proxies to inform on cold bleaching susceptibilities, lies in the combination of both heat and cold CBASS allowing to resolve the viable temperature range of coral species. As such, we propose that

		warm the tropical waters where corals are found? Technically, this study is interesting, but I am not sure if the ability to determine cold bleaching thresholds is mission critical for the conservation and protection of corals into the warmer futures that they will experience. The writing is clear and well-explained (some minor comments below). I did find the Discussion a bit verbose and I believe the text could be shortened. Overall, this is a valuable study that was well thought out and performed. The statistics are appropriate and it is clear that the research team was well equipped to conduct this study and analyses.		species with a wider viable temperature range may be tolerant towards heat stress due to their ability to withstand and benefit during the critical time for recovery (i.e., winter). Additionally, corals that experienced temperature stress in summer may be more susceptible to cold stress in following winter months, or vice versa. We have clarified this in the main manuscript. Edits: 1.504-511: “Determining heat and cold temperature tolerance thresholds, and recovery potentials allows resolving the viable temperature range of corals, with implications for our understanding and prediction of climate change consequences in coral reefs. The results from our study highlight that all investigated coral species are prone to both cold and heat-stress induced bleaching, with measuring the viable temperature ranges of coral species offering an approach to assess climate resilient reefs. Importantly, combining hot and cold CBASS assays allows resolving the full viable temperature range of corals, highlighting that species with broader ranges may be more resilient to heat stress by withstanding and recovering during subsequent cold events¹¹². Winter bleaching events triggered by severe weather anomalies or cold-water upwelling can pose threats to corals comparable to those caused by heat stress events.”
3	32	there is an extra comma after sp.	Edited.	We agree that this is confusing. We slightly re-arranged the sentence to avoid this. Edits: 1.32: “...in Acropora sp. during summer.”
4	55	comma not needed after "temperature"	Edited.	Agreed, thank you.
5	81	extra space before reference citation	Edited.	Agreed, thank you.
6	132	The presentation of the bleaching thresholds to such a fine decimal is tricky since you only tested stress at 3 degree intervals. Perhaps this precision has already been demonstrated using the CBASS	Noted but not changed.	We appreciate the reviewer’s suggestion. We carefully disagree, however, as 2 decimal digits is a standard, particularly considering that the reported ED50 values are model estimates.

		system, but that is some real fine scale precision and I wonder if it might be more fair to round those numbers a bit.		These estimates are precise and expressed as mean with their confidence being expressed as \pm standard error.
7	224/Figure 3	Figure 3, it is very challenging to distinguish between the various shades of colors on this bacterial abundance plot. Please consider changing the color ranges.	Edited.	We agreed with the reviewer, we therefore adjusted the color palette used for Figure 3.
8	336	Is this 6-9°C temperature range shown here linked to the natural temperature range in the environment by season for these corals?	Edited.	We appreciate the question asked by the reviewer. We refrained from direct comparisons from bleaching thresholds obtained here to the field, due to the general limitations of the CBASS assay. We, however, addressed the point raised by the reviewer by highlighting that the observed high cold (6-9°C) and relatively small heat ranges (1-3°C) may be attributed to the fact that corals live close to their upper thermal tolerance limits. Edits: 1.357-361: “This suggests natural acclimatization to naturally fluctuating temperatures with species-specific response patterns to both heat and cold temperature stress. Additionally, these findings complement that tropical reef-building corals live close to their upper thermal tolerance limits, such that even small (~1–3 °C) summer anomalies can trigger bleaching^{80,81}.”
9	359	What about the microbiome of the surrounding seawater? If some of these coral species are known to have microbiomes influenced by the environment, could the changes noted here be due to changes in the microbial composition in seawater? I don't remember seeing anything about sampling of the water around the corals.	Edited.	We agree with the point raised by the reviewer. However, we did not include seawater microbiome analyses in the present study. Nevertheless, we addressed this point in the discussion referring to a study by Lima et al. (2021, Microbial Ecology). Edits: 1.390-394: “Potentially, such shifts in the coral microbiome may also be influenced by the surrounding seawater microbial community, which was not assessed in this study. Previous work has shown that environmental microbiota can seed or shape coral-associated bacterial assemblages⁸⁸. Thus, part of the variation we observed may reflect concurrent changes in the ambient seawater microbiome with species-specific responses.”

10	428	change to "Here, we"	Edited.	Agreed, thank you.
11	447	Did you see any direct damage to the chloroplasts as a result of the cold stress as seen in studies like Bieri et al 2016? They did use 4°C cold shock but the cold has been reported to cause severe damage to the photosystem.	Edited.	We appreciate this interesting point raised by the reviewer. Even though we did not assess chloroplast damage in the present study, we addressed this point using the suggested reference in the discussion. Edits: 1.488-492: “Even though we did not assess potential structural damage to chloroplasts in the present study, Bieri et al. (2016)¹⁰⁸ reported that cold stress can impair chloroplast integrity and function in corals, suggesting that the enhanced photosynthetic efficiency we observed following a recovery phase may reflect compensatory mechanisms at the cellular or holobiont level rather than the absence of cold-induced damage.”
12	522	Perhaps clarify that the stressed temperatures were 24, 21, 18, 15°C?	Edited.	Agreed, has been clarified.